# A direct neural signature of serial dependence in working memory

Cora Fischer[1,2], Jochen Kaiser[1,2], Christoph Bledowski[1,2]*

[1]Goethe University Frankfurt, Institute of Medical Psychology, Frankfurt am Main, Germany; [2]Goethe University Frankfurt, Cooperative Brain Imaging Center, Frankfurt am Main, Germany

## eLife Assessment

This study reveals a neural signature of a common behavioural phenomenon: serial dependence, whereby estimates of a visual feature (here motion direction) are attracted towards the recent history of encoded and reported stimuli. The study provides **solid** evidence that this phenomenon arises primarily during working memory maintenance. The pervasiveness of serial dependencies across modalities and species makes these findings **important** for researchers interested in perceptual decision-making across subfields.

*For correspondence: bledowski@em.uni-frankfurt.de

**Abstract** Serial dependence describes the phenomenon that current object representations are attracted to previously encoded and reported representations. While attractive biases have been observed reliably in behavior, a direct neural correlate has not been established. Previous studies have either shown a reactivation of past information without observing a neural signal related to the bias of the current information, or a repulsive distortion of current neural representations contrasting the behavioral bias. The present study recorded neural signals with magnetoencephalography (MEG) during a working memory task to identify neural correlates of serial dependence. Participants encoded and memorized two sequentially presented motion directions per trial, one of which was later retro-cued for report. Multivariate analyses provided reliable reconstructions of both motion directions. Importantly, the reconstructed directions in the current trial were attractively shifted toward the target direction of the previous trial. This neural bias mirrored the behavioral attractive bias, thus reflecting a direct neural signature of serial dependence. The use of a retro-cue task in combination with MEG allowed us to determine that this neural bias emerged at later, post-encoding time points. This timing suggests that serial dependence in working memory affects memorized information during read-out and reactivation processes that happen after the initial encoding.

## Introduction

Our visual input can change quickly from moment to moment. Nevertheless, we represent our environment as remarkably stable and coherent across short periods of time. This coherence is thought to be supported by serial dependence (*Fischer and Whitney, 2014*), that is, the phenomenon that a current visual object is reported as more similar to a previously encountered visual object than it actually was. This attractive bias acts most strongly between current and previous objects that are close in time and space, that are attended, and that are similar to each other both with regard to the visual feature that has to be recalled and with regard to accompanying context features (*Czoschke et al., 2019*; *Fischer et al., 2020*; *Fischer and Whitney, 2014*). Furthermore, serial dependence occurs at

different levels of object processing including encoding (*Murai and Whitney, 2021*), memorization (*Bliss et al., 2017*), and read-out or decision (*Fritsche et al., 2017*; *Pascucci et al., 2019*).

While the cognitive aspects of serial dependence have been intensively studied (for reviews see *Kiyonaga et al., 2017*; *Pascucci et al., 2023*), its neural mechanisms are still largely unclear and a matter of active research. One important aspect that has been investigated on the neural level is the nature of the memory traces that impact current processing and the timing of their reactivation. Single-unit recordings from monkey prefrontal cortex combined with electroencephalography (EEG) and transcranial magnetic stimulation in humans have demonstrated that information about a previous stimulus was reactivated shortly before a new stimulus was encoded (*Barbosa et al., 2020*). This study found that the strength of this pre-encoding reactivation in the prefrontal cortex predicted the attractive bias of the behavioral response to the new stimulus. Moreover, subjects showed an enhanced serial bias after single-pulse transcranial magnetic stimulation of their prefrontal cortex, which further supported a causal link between the observed reactivation and serial dependence. Another study (*Akrami et al., 2018*) showed that memory traces from previous trials recorded from the posterior parietal cortex in rats reappeared during the inter-trial interval (ITI) of a working memory task, were present throughout the next trial, and predicted the behavioral bias. The observed results indicate that frontal and parietal regions are involved in the storage of memory traces and their transfer into current processing. Several human EEG studies have also shown reactivation signatures of the previous stimulus or a response during the current trial, establishing reactivation as an important neural prerequisite for the emergence of serial dependence (*Bae and Luck, 2019*; *Barbosa et al., 2020*; *Fornaciai and Park, 2018*; *Fornaciai and Park, 2020*; *Fornaciai et al., 2023*; *Luo and Collins, 2023*; *Ranieri et al., 2022*).

Rather than looking for reactivation of past information, recent studies have investigated the neural activity elicited by the current stimulus itself. This approach provides a more direct measure of how the current neural representation is influenced by the recent past. Neural activity in the visual cortex measured with functional magnetic resonance imaging (fMRI) was influenced by orientations presented on a previous trial during the encoding and maintenance of a current orientation (*St. John-Saaltink et al., 2016*). However, as the current orientation was either identical or differed by exactly 90° from the previous trial, it remained unclear whether this neural bias reflected an attraction or repulsion in relation to the past. Furthermore, the bias in the V1 signal was partially explained by the orientation that was presented at the same position in the previous trial, which could reflect a reactivation of the previous orientation rather than an altered current orientation. In a more recent fMRI study (*Sheehan and Serences, 2022*), the current neural representation of an orientation in the visual cortex was repulsed from an orientation remembered in the previous trial, which was in contrast to an observed behavioral attraction. In line with this finding, a recent magnetoencephalography (MEG) study (*Hajonides et al., 2023*) observed a neural repulsion of a current representation during encoding, in contrast to a behavioral attractive bias toward past information. The authors of both studies proposed that the attractive behavioral bias is driven by post-encoding processes. This view is supported by another computational model (*Fritsche et al., 2020*), which states that early visual representations are repulsed and the behavioral attraction emerges during post-encoding read-out of memorized information.

Until now, a direct reflection of the attractive bias of current information on the neural level has not been observed, and it has remained unclear during which processing step memory representations are actually biased toward the past. The current study applied an inverted encoding model (IEM) analysis to neural signals measured with MEG to reconstruct the information about currently remembered stimuli (the term 'reconstruction' or 'reconstructed representation' refers to the reconstructed channel response, i.e., the output of the IEM analysis; it is not supposed to relate to the reconstruction of single-unit responses and is used throughout the manuscript for brevity and readability). We expected that a direct neural signature of serial dependence should mirror the attractive bias in behavior by showing an attractive shift of the reconstructed information toward the target information in the previous trial. The aforementioned models *Fritsche et al., 2020*; *Sheehan and Serences, 2022* have assumed that neural representations become attracted toward the recent past after encoding. We employed a task that allowed us to differentiate between early processes including encoding and maintenance and later, post-encoding read-out processes to inform these models and determine not only if, but also during which processing stage the neural representation becomes attracted toward

**Figure 1.** Experimental paradigm and behavioral results. (**A**) Participants had to remember two motion directions (S1 and S2) per trial (motion directions are indicated here by gray arrows for illustration only). The stimuli were either red or green. After a short delay, a retro-cue indicated which of the two motion directions (red or green) they had to recall by adjusting a randomly oriented line. (**B**) Response errors in the current trial were biased toward the target direction of the previous trial (pink curve), but not toward the previous non-target direction (gray curve). The x-axis shows the motion direction difference between the previous and the current target, with positive values indicating that the previous target direction was more clockwise than the current target direction. The response error on the current trial is displayed on the y-axis, with positive values indicating a clockwise deviation from the true target direction. Therefore, shifts of the response errors to the lower left and upper right quadrant indicate an attraction, whereas shifts to the upper left and lower right quadrant indicate a repulsion. The asterisk marks significant attractive serial dependence toward the previous target direction (one-sided permutation test, N = 10).

the recent past. More specifically, we asked subjects to encode two motion directions and memorize them for a short delay, followed by a retro-cue that indicated which direction served as the target for an upcoming report. This retro-cue design, in combination with the high temporal resolution of MEG, enabled us to track the current representations across the whole trial to determine not only if, but also during which processing stage the neural representations become attractively biased toward the direction reported in the previous trial.

## Results

### Behavioral results

Ten human participants (seven females) memorized two sequentially presented visual objects per trial (S1 and S2), while their neural signals were measured via MEG. Each participant completed two sessions with a total of 1022 trials. The objects were colored dot fields, and participants had to remember the motion direction of the dots. After a short delay, one of the motion directions was cued via the dot color (red or green, *Figure 1a*). As expected, we observed a serial dependence toward the target of the previous trial, as the reported target motion direction of the current trial was systematically attracted to the retro-cued motion direction of the previous trial. This attraction (*Figure 1b*) followed a derivative-of-Gaussian (DoG)-shaped curve with an amplitude parameter of 3.51° (bootstrapped SD: 0.479°, lower 95% of permutations between –3.51° and 2.03°, p < 0.001, $R^2 = 0.236$) and a *w* parameter of 0.024, equaling a width of 47.37° (full width at half maximum, FWHM). We ran one-sided permutation tests (*n* = 10) to assess whether this attraction differed from zero. In contrast, we did not observe a significant serial dependence toward the non-target of the previous trial (amplitude: –1.02°, bootstrapped SD: 0.788°, middle 95% of permutations between –1.63 and 1.49, p = 0.206, $R^2 = 0.014$; *w* parameter: 0.046, equaling 24.53° FWHM), which replicated our previous observations (*Czoschke et al., 2019*; *Fischer et al., 2020*).

We also estimated serial dependence between the current and the previously cued object with regard to their color congruence. Serial dependence was modulated by task-relevant color. It was stronger when the current item had the same color as the cued item of the previous trial (amplitude = 4.02°, SD = 0. 528°, lower 95% of permutations between –3.70° and 2.41°, p < 0.001, $R^2 = 0.167$) than when their colors differed (amplitude = for all three epochs, we observed 2.99°, SD = 0. 538°, lower 95% of permutations between –2.99° and 1.79°, p = 0.001, $R^2 = 0.106$). The amplitude difference amounted to 1.02°, p = 0.038. This replicated the context effect found in our previous study (*Fischer et al., 2020*). The behavioral results thus showed that our two-item working memory task, performed during an MEG recording, produced reliably an attractive bias between target items across trials.

### MEG results

We applied an IEM (*Brouwer and Heeger, 2009*; *Brouwer and Heeger, 2011*; *Sprague and Serences, 2013*) to the MEG data from all active sensors (271) within defined time windows of 100

ms length. MEG data were recorded in two sessions on different days. Specifically, we constructed an encoding model with 18 motion direction-selective channels. Each channel was designed to show peak sensitivity to a specific motion direction, with gradually decreasing sensitivity to less similar directions. In a training step, the encoding model was fitted to the MEG data of one session to obtain a weight matrix that indicates how well the sensor activity can be explained by the modeled direction. In the testing step, the weight matrix was inverted and applied to the MEG data of the other session, resulting in a response profile of 'reconstruction strengths', that is, how strongly each motion direction was present in a trial. When a specific motion direction was present in the MEG signal, the reconstruction strengths peaked at that specific direction and decreased with increasing direction difference. If no information was present, reconstruction strengths were comparable across all modeled directions, that is, the response profile was flat. To integrate response profiles across trials, single-trial profiles were aligned to a common center direction (i.e., 180°) and then averaged.

To quantify the accuracy of each IEM reconstruction, that is, how well the response profile represents a specific motion direction relative to all other directions, we computed the 'reconstruction fidelity'. Fidelity was obtained by projecting the polar vector of the reconstruction at every direction angle (in steps of 1°) onto the common center (180°) and averaging across all direction angles (*Rademaker et al., 2019*; *Sprague et al., 2016*). As such, 'reconstruction fidelity' is a summary metric with fidelity greater than zero indicating an accurate reconstruction (see Methods for details).

## Reconstruction of the remembered directions per epoch

First, we estimated how well we could reconstruct the motion direction of both current items throughout the current trial. For this aim, we divided the current trial into three epochs, including the encoding and maintenance epochs of S1 and S2 (termed S1 and S2 epochs, respectively) and the retro-cue epoch. Specifically, during the S1 epoch, we reconstructed the direction of S1, during the S2 epoch, we reconstructed the direction of S2, and during the retro-cue epoch, we reconstructed the motion direction of the currently retro-cued S1 or S2 (current target). Our reconstruction procedure used the stimulus information during the training step by modeling the presented or memorized directions with the chosen encoding model and fitting them to the MEG data from the corresponding epoch in one MEG session by creating a weight matrix. Then the inverted weight matrix was applied to independent testing MEG data from the corresponding epoch in the other MEG session to reconstruct the direction information, which was then aligned on a single-trial level to a common center with regard to the presented or memorized direction. For example, to reconstruct S1 direction during the S1 epoch, the model was trained on data from the S1 epoch in one MEG session and applied to the S1 epoch of the other MEG session (see Methods for details).

For all three epochs, we observed successful reconstructions of the corresponding motion directions with a fidelity significantly greater than zero (S1 epoch: p < 0.002, cluster extent: 300–1200 ms, mean fidelity = 0.139; S2 epoch: p < 0.002, cluster extent: 0–1300 ms, mean fidelity = 0.120; retro-cue epoch: p < 0.002, cluster extent: 300–1500 ms, mean fidelity = 0.115) (*Figure 2a, b*). This shows that the information about the motion direction of S1, S2, and the retro-cued target was represented in the neural signals during the respective epochs.

For completeness, we also reconstructed the direction of S1 during the S2 epoch, as the direction of S1 was still maintained in working memory in parallel to the concurrent encoding and maintenance of the S2 direction (*Figure 3a, b*). Here, we used the MEG data from the S2 epoch again, but now for S1 training, that is, the model was informed about S1 direction. Accordingly, the alignment to a common center direction during testing was done with regard to the S1 direction. We observed that during the S2 epoch, the direction of S1 could be successfully reconstructed with a fidelity significantly greater than zero (S1 during S2 epoch: p = 0.005, $p_{Bonferroni}$ = 0.015, cluster extent: 100–700 ms, mean fidelity = 0.065). However, its mean fidelity within the significant cluster was significantly smaller than the fidelities of the currently relevant direction during all other epochs, that is, S1 during S1 epoch, S2 during S2 epoch, and target during the retro-cue epoch (all p-values below 0.005).

Furthermore, we tried to reconstruct the motion direction that was not retro-cued during the retro-cue epoch, that is, the non-target of a given trial that became task-irrelevant after the retro-cue. As expected, the information about the non-target item could not be reconstructed in the retro-cue phase (all cluster p-values >0.05).

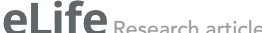

**Figure 2.** Reconstruction, fidelity, and systematic shifts of currently relevant motion direction. (**a**) The reconstructed direction of the currently relevant motion direction across the course of a trial. The left panel shows the reconstruction of the S1 direction, the middle panel the reconstruction of the S2 direction, and the right panel the reconstruction of the cued direction (either S1 or S2). Higher reconstruction strength is depicted in yellow and lower strength in dark blue. Time points (*x*-axis) with direction information show a pronounced yellow area around the common center set to zero (*y*-axis) and darker areas with increasing distance from this center. (**b**) Fidelity of reconstructed direction of the currently relevant motion direction across the course of a trial. The left panel shows the fidelity of the S1 direction, the middle panel depicts the fidelity of the S2 direction, and the right panel shows the fidelity of the cued direction (either S1 or S2). Small circles show the fidelity of each participant. Colored circles indicate time points with a fidelity significantly greater than zero (cluster-based permutation test within each epoch, *N* = 10). (**c**) The reconstructed direction during all time points with a significant reconstruction fidelity was averaged within each epoch, and the maximum of the reconstructed direction was compared to the common center set to zero. A negative shift of the reconstructed mean indicates a repulsion from the target direction of the previous trial, whereas a positive shift indicates an attraction. For a more detailed visualization, the range of direction shifts from –40° to +80° is magnified, and reconstructed shift values for each participant are indicated by dots. The upper row shows the bootstrapping distributions (see section on MEG analysis for details). Significant shifts are marked with an asterisk (one-sided bootstrapping test, *N* = 10).

## Attractive shift of motion direction reconstructions as a direct signature of serial dependence

The main analysis of our study aimed to identify the neural signature of serial dependence. For this aim, we looked for an attractive bias of the reconstructed neural representations that mirrored the

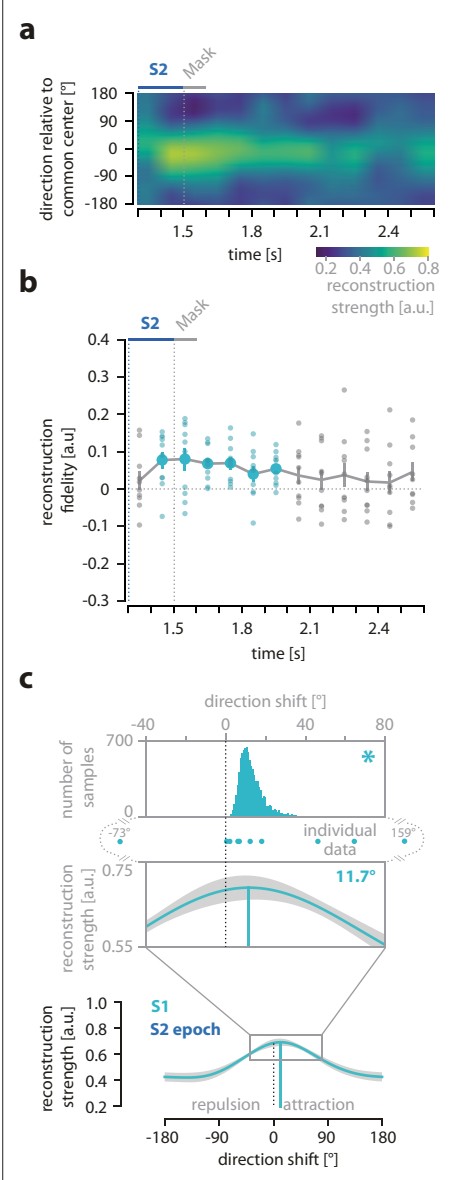

**Figure 3.** Reconstruction, fidelity, and systematic shifts of S1 direction during the S2 epoch. (**a**) The reconstructed direction of the S1 motion direction during the S2 epoch. (**b**) Fidelity of reconstructed S1 motion direction across the course of the S2 epoch. Small circles show the fidelity values for each participant. Colored circles indicate time points with a fidelity significantly greater than zero (cluster-based permutation test within each epoch, N = 10). (**c**) The reconstructed direction for all time points with a significant reconstruction fidelity was averaged within the S2 epoch to indicate attractive and repulsive shifts in relation to the target of the previous trials. Significant shifts are marked with an asterisk (one-sided bootstrapping test, N = 10). For details, refer to *Figure 2*.

attractive behavioral bias by testing whether the mean of the significant reconstructions within the S1, S2, or retro-cue epochs showed positive shifts of their maxima from the common center of 180° toward the previous target (*Figure 2c*; see Methods for details). We found that the shift of the mean reconstruction of the first and second item did not differ from 0° for the S1 epoch (–2.83°, p = 0.743, bootstrapping test, $p_{FDR}$ = 0.870, 95% of permutations between –11.60° and 6.00°) or for the S2 epoch (–2.50°, p = 0.870, bootstrapping test, $p_{FDR}$ = 0.870, 95% of permutations between –7.57° and 2.11°), respectively. In contrast, for the retro-cue epoch, we found that the mean reconstruction of the current target was shifted toward the previous target (10.22°, p = 0.009, bootstrapping test, $p_{FDR}$ = 0.026, 95% of permutations between 1.64° and 21.97°; *Figure 2c*). In addition, there was an attractive shift of the S1 direction toward the target of the previous trial during the S2 epoch (11.69°, p < 0.001, bootstrapping test, 95% of permutations between 4.96° and 31.06°; *Figure 3c*). Thus, our MEG data provided evidence for an attractive distortion of current neural representations toward the target information in the previous trial, thereby revealing a direct neural signature of serial dependence.

## Correlation between neural representation and behavioral response

To assess whether the observed item reconstructions from the MEG signals were behaviorally relevant, we correlated them with the subjects' behavioral responses. Specifically, we calculated the circular correlation between the maximum of the stimulus direction reconstructions and the response error for each time point and epoch separately across *single trials*, that is, we tested whether the shift/deviation of the neural response from the true value, for example, in the clockwise (CW) direction, was accompanied by a corresponding shift/deviation of the behavioral response.

Indeed, *Figure 4* shows that the reconstructed shift of the retro-cued target item during the retro-cue phase predicted the upcoming behavior between 400 and 800 ms after the onset of the retro-cue (p = 0.003, permutation test, $p_{Bonferroni}$ = 0.009; mean r = 0.030). In contrast, there were no significant correlations for any other time points during the S1 and S2 epochs (cluster-based permutation test exact test, one-tailed). While the maximum of the S1 reconstruction also showed a significant shift toward the previous target, there

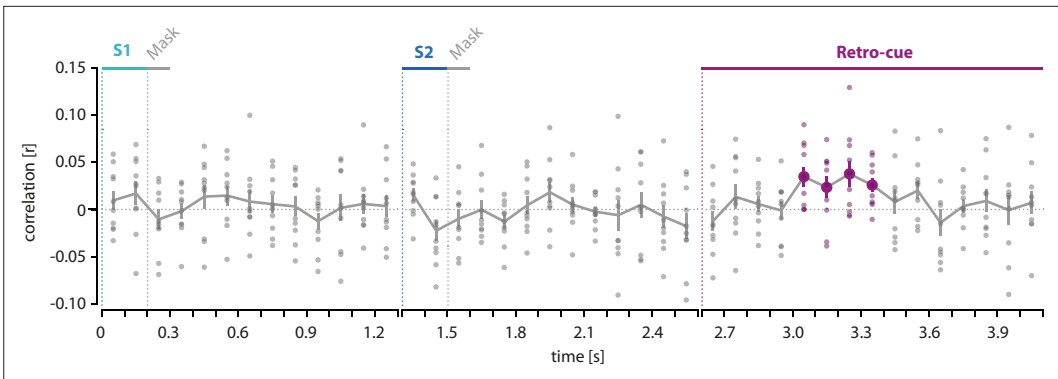

**Figure 4.** Correlation between single-trial shift of reconstruction and upcoming response. For each time point, the shift of the maximum of the reconstructed direction in a single trial was correlated with the response deviation from the target direction. Small circles show the *r*-values of each participant. Time points where the shift of the reconstructed direction significantly correlated with the upcoming response deviation are indicated by colored circles (cluster-based permutation test within each epoch).

were no significant correlations of the S1 reconstruction with the response error during the S2 epoch (no clusters with a p-value <0.05).

For each time point, the shift of the maximum of the reconstructed direction in a single trial was correlated with the response deviation from the target direction. Small circles show the *r*-values of each participant. Time points where the shift of the reconstructed direction significantly correlated with the upcoming response deviation are indicated by colored circles (cluster-based permutation test within each epoch).

The presented single trial-wise correlation between the neural shift and behavioral shift did not directly incorporate the behavioral bias toward the previous trial. Thus, in order to relate the neural attractive shift and the behavioral indicators of serial dependence more directly, we performed

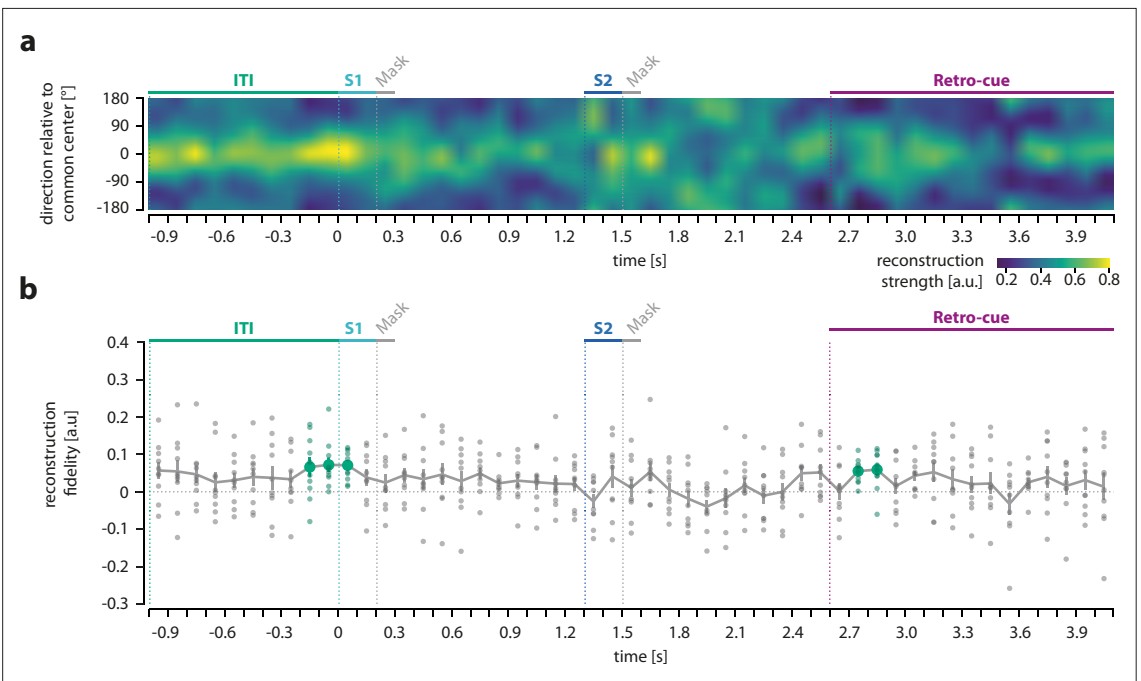

**Figure 5.** Reconstruction and fidelity of previous target direction. (**a**) The target direction of the previous trial was reconstructed throughout the last second of the inter-trial interval (ITI) and the whole following trial. (**b**) Fidelity of reconstructed direction of the previous target. Small circles show the fidelity of each participant. Colored circles indicate time points with a fidelity significantly greater than zero (cluster-based permutation test within each epoch, *N* = 10). For details, refer to *Figure 2*.

an additional correlation analysis on aggregated data on the *between-subject* level. In detail, we correlated the individual neuronal shift during the retro-cue epoch with the two individual parameter fits of the behavior shift, that is, amplitude (*a*) and tuning width (*w*). In line with the correlation analysis on single-trial level, the analysis on the *between-subject* level also revealed a significant correlation between the *w* parameter of serial dependence and the neural shifts for the retro-cue epoch. Details of this correlation analysis are displayed in Appendix 1.

### Reactivation of the previous target direction

Finally, we used our IEM analysis approach to replicate previous results that have shown a reactivation of memory traces from the previous trial before the stimulus presentation of the current trial (*Barbosa et al., 2020*). Specifically, we reconstructed the motion direction of the target from the previous trial during the 1 s of the ITI immediately preceding the current trial (*Figure 5*). Consistent with previous findings from monkey single cell and human EEG recordings (*Barbosa et al., 2020*), we observed that the information about the previous target was reactivated just prior to the onset of the next trial. Direction reconstructions showed a fidelity significantly greater than zero during a time period at the end of the ITI (p = 0.019, cluster extent: –200 to 100 ms (relative to S1 onset), mean fidelity = 0.069). Importantly, we also tracked the information of the previous target throughout the whole current trial to test whether it was reactivated also during the processing of the current items. Interestingly, we found that the target from the previous trial was reactivated once more, but at a later time period during the current trial, that is, when the retro-cue indicated which item of the current trial was the target for the upcoming response (p = 0.040, cluster extent: 2700–2900 ms relative to trial onset, mean fidelity = 0.057). In contrast, we found no evidence of previous target reactivations during the encoding or maintenance of current items (no clusters with a p-value <0.05).

Additionally, in order to test whether a reactivation of the previous target could explain the neural attractive shift observed in the current trial, we reconstructed the motion direction of the previous target from MEG data recorded during the different epochs of the current trial. Specifically, we trained the encoding model either on the presented stimuli in the S1 and S2 epochs or on the currently relevant item during the retro-cue epoch, respectively, and tried to reconstruct the target direction of the previous trial within each epoch. We observed that there was no cross-reconstruction between the current and the previous representation during the retro-cue epoch, which indicates that the observed attractive shift of the current target reconstruction following the retro-cue was not driven by a reactivation of the previous target in the same neural code. However, during the beginning of the S2 epoch, we observed a significant cross-reconstruction, that is, previous target reactivation co-occurred with the reconstruction of S2 while it was presented to the subjects. The complete analysis and results are described in Appendix 1.

### Eye movement control

To control for influences of eye movements on the MEG data, we recorded horizontal and vertical electrooculograms to identify blinks and eye-movement-related independent components of the MEG data that were removed during our MEG preprocessing (see Methods). In addition, we also recorded continuous gaze position with a higher spatial resolution using an eye tracker. We used the eye-tracking data to estimate whether there was a systematic relationship between eye movements and the motion direction of currently processed items in the S1, S2, or retro-cue phases. We found that gaze directions were systematically related to presentation and memorization of the stimulus direction (S1 and S2 epoch) and its cue-based retrieval (retro-cue phase). They also varied considerably between subjects. Most importantly, gaze directions were not systematically related to the MEG data, thus ruling out that they had mainly driven our MEG results. The details of the eye movement control analysis are displayed in Appendix 1. Importantly, the central finding of our study is that serial dependence emerged at a later post-encoding stage of object processing in working memory, irrespective of any possible eye movements.

## Discussion

The current study aimed to identify a direct neural signature of serial dependence. Previous studies searching for neural markers of this attractive bias have investigated either the reactivation of

information from the past trial in the current trial or have tried to directly capture biases of current item representations by information from the past. The present study found evidence for both types of neural correlates. We replicated reactivations of target representations of the previous trial. More importantly, our study provides the first evidence for an attractive bias of neural representations in the current trial toward information from the previous trial, which mirrors the behavioral effects of serial dependence and as such represents a neural signature of serial dependence.

The reactivation of past information in the current trial is crucial because an attractive bias can only occur when information from the past has left a memory trace that interacts with the processing of the current object. *Barbosa et al., 2020* have shown that the working memory trace of an object representation from a previous trial, measured via decoding of both neural activity in monkey prefrontal cortex and human EEG signals, was reactivated just before new visual input was presented in the current trial. This was in line with modulations of early visual evoked potentials in human EEG by past stimulus information found in numerosity tasks (*Fornaciai and Park, 2018*; *Fornaciai and Park, 2020*). Using human EEG, *Bae and Luck, 2019* observed the reactivation of past stimulus information also during encoding and maintenance phases in the current trial of an orientation working memory task. In our study, we applied a similar analysis approach and tracked the reconstructed motion direction of the target item from the previous trial both during the ITI, that is, prior to the encoding of the new stimulus, and throughout the whole current trial. Similar to the previous studies, we also found that the memory trace of the previous target was reactivated just prior to the onset of the current trial and later in the current trial after the retro-cue indicated the current target.

Reactivations of information from the previous trial may be seen as crucial but indirect neural correlates of serial dependence. A memory trace of a previously relevant representation is an important prerequisite for the emergence of serial dependence, since a current representation can only be biased by integrating the information from the past into the present trial. However, our main goal was to observe the direct effect of this past information on current representations. More specifically, we asked whether neural representations of current information are biased toward the recent past in the same way as the behavioral response. Previous studies that aimed to measure neural correlates of serial dependence directly found repulsive rather than attractive distortions of neural representations in relation to the recent past, which were in contrast to the attractive bias in the behavioral response (*Hajonides et al., 2023*; *Sheehan and Serences, 2022*). The repulsive neural biases were attributed to visual adaptation during early perceptual processes. In contrast to these previous reports, the present study found attractive biases of representations in the present trial toward information from the previous trial that were consistent across the neural and behavioral level.

We propose that our combination of a retro-cue paradigm with the high temporal resolution of MEG accounts for the successful demonstration of serial dependence at the neural level in the present study. *Sheehan and Serences, 2022* recorded fMRI while participants made a delayed report about a single memorized grating orientation. In contrast to the attractive behavioral bias, the orientation reconstructed from the fMRI signals measured in the visual cortex was repulsed away from the orientation presented in the previous trial. The authors proposed a computational model in which early encoding-related signals are repulsed from the recent past. The attraction observed in behavior was attributed to a later decoding process that overcomes the repulsion of sensory signals and leads to serial dependence but is not directly detectable in hemodynamic response patterns. This is consistent with the notion of an ideal observer model that combines two steps of object processing, including an early encoding and a later decoding of current information to explain serial dependence (*Fritsche et al., 2020*). According to this model, a repulsive bias occurs early during encoding, whereas an attractive bias is due to the read-out of the current object representation in a Bayesian history-dependent manner. Previous behavioral studies have already identified read-out as a crucial processing stage for serial dependence. *Fritsche et al., 2017*, for example, asked participants either to remember and report the orientation of a grating after a short delay (working memory task) or to compare two simultaneously presented orientations directly (perception task). They found a repulsive bias in the perception task, when the grating was still visually present and compared to another grating, whereas in the working memory task, the current target orientation was attracted toward the orientation reported in the preceding trial. This supported the emergence of serial dependence at a late, post-perceptual processing stage.

Similarly, *Hajonides et al., 2023* observed a repulsion of currently remembered orientations as decoded from MEG signals, while the behavioral recall was attracted toward the past. Participants performed a working memory task with two sequentially presented sample stimuli similar to our study. However, instead of using a temporally separated retro-cue, *Hajonides et al., 2023* presented a cue that indicated the target for later recall simultaneously with the second sample stimulus. As participants knew which item would be the response target as soon as the second item was presented, the read-out process for a specific item could not be disentangled from the encoding or maintenance stages. Multivariate analysis of MEG signals revealed that the reconstructed orientation information during and shortly after stimulus presentation was repulsed from the orientation recalled in the previous trial. In line with *Sheehan and Serences, 2022*, the authors assumed that the observed repulsion at the neural level reflected visual adaptation, whereas the attraction of the neural representation may have occurred at later processing stages, which were not accessible by their analysis. *Hajonides et al., 2023* speculated that the integration of past memory traces, putatively stored in parietal areas (*Akrami et al., 2018*), only happened during the behavioral response. This would mean that the neural representation of the current representation is not affected by serial dependence, but that this bias emerges as a result of integration processes during the response.

The present study provided support for the notion that the neural markers of serial dependence occur during post-perceptual phases of a working memory task, which is in agreement with existing models and behavioral findings (*Fritsche et al., 2020*; *Hajonides et al., 2023*; *Sheehan and Serences, 2022*). The use of a retro-cue task, which facilitates the segregation of earlier and later processing stages, in combination with motion direction reconstructions from MEG signals, revealed shifts of the current representation toward the motion direction of the target in the previous trial. We observed such an attractive bias only at post-encoding processing stages in two time windows: First, for S1 during and after the presentation of S2 and, second, for the cued target stimulus after the presentation of the retro-cue. This demonstrates that serial dependence can be observed on the neural level as a distortion of current neural representations that mirrors the attractive bias in behavior. This direct neural signature of serial dependence emerges after encoding but prior to the response, thus speaking against the proposal that serial dependence is attributable to response-related processes (*Hajonides et al., 2023*).

To further underscore the relevance of the observed neural distortions, we examined whether the reconstructed neural representation predicted the behavioral response deviation. Indeed, we observed a positive correlation between the deviations of the neural reconstruction and response errors on a single-trial level. Interestingly, this correlation was found during the presentation of the retro-cue, coinciding with the time point at which the neural representation of the current target stimulus was biased toward the recent past. This observation was further supported by a positive correlation between the attractive shift of the neural reconstruction during the retro-cue epoch and a behavioral measure of serial dependence on the between-subject level. In addition, we observed a reactivation of previous target information during the retro-cue presentation. The distortion of the current neural representation could thus be the result of an integration of past information. Taking these findings together, the retro-cue period emerges at the time point when different neural markers of serial dependence appear together. This suggests that the selective read-out of memorized information for an upcoming report might be a crucial process for serial dependence during working memory.

The neural representation of S1 already became attracted toward the target of the previous trial when S2 was presented. Notably, the reconstruction of the S1 direction had dropped to baseline and reappeared during the S2 epoch. This fits well with the observation that memory content that has become neurally silent can be successfully reconstructed from the impulse response elicited by a 'ping' or probe stimulus (*Wolff et al., 2015*; *Wolff et al., 2017*). This impulse-driven reconstruction is thought to reflect a 'hidden state' of working memory, in which unattended information is stored. Therefore, the observed S1 reconstruction elicited by S2 presentation in our study likely reflects the same type of memory read-out, which might have driven the observed neural distortion. On the other hand, S1 had been maintained for a certain time when S2 was presented. As *Bliss et al., 2017* observed, serial dependence in behavior increases as a function of delay duration, so the distortion of the neural representation of S1 during the S2 period could also reflect an increasing bias during working memory maintenance.

Several recent studies have shown that serial dependence operates already during perception (*Manassi et al., 2018*; *Murai and Whitney, 2021*). In the current study, the neural representation of the current information became biased only after encoding. However, we did observe a reactivation of previous information as an indirect marker of serial dependence before and during the onset of S1, replicating previous results (*Barbosa et al., 2020*). This early reactivation of past information could reflect a comparison of prior information with new visual input. *Cicchini et al., 2021* proposed a predictive coding framework, in which priors informed by past visual input are compared against incoming visual information during early encoding. This comparison results in a prediction error, which determines whether a current stimulus will be biased toward the past. In combination with the observed distortion of the neural representation, this could indicate a two-step process: New visual information is compared against past information during early encoding processes, and the integration of past and current information happens during later processing stages, before the actual report is given.

How can these findings of serial dependence operating already during perception (*Fischer and Whitney, 2014*; *Manassi et al., 2018*) be reconciled with other behavioral studies and our findings that pinpoint the emergence of serial dependence at post-perceptual processing stages (*Bliss et al., 2017*; *Fritsche et al., 2017*)? One possibility is that serial dependence does not operate at a single but rather at different levels of object processing, including early perceptual and later, memory-related processes (*Whitney et al., 2022*). Moreover, perception and working memory might not be as different as they appear: both functions build mental representations from sensory input, involving feedforward and feedback processing. Hence, encoding and decoding of information, including the neural computations in lower- and higher-level cortical areas, represent integral parts of both perception and working memory. Thus, in a perceptual task that requires an immediate response when visual input is still physically present, read-out of the noisy information via decoding is closely intertwined with its encoding. In working memory tasks that require encoding and memorization of several items, the read-out of a single item is typically postponed: It starts when a retro-cue or probe indicates which memorized item is task-relevant. Serial dependence that operates during the read-out stage of object processing appears to happen both early in perception and later in working memory tasks. However, as shown in this study, in order to identify the neural basis of serial dependence, a multi-item working memory task offers the advantage of separating different processing stages to isolate the time point when an object representation is actually distorted. Furthermore, while the present results indicate a particular importance of post-encoding processes for serial dependence in working memory, they do not suggest that serial dependence could not emerge during encoding. First, we cannot exclude that our data analysis was not sensitive enough to detect small shifts in the neural signals during encoding, especially as our analysis involved averaging across time points to increase the signal-to-noise ratio. Second, with the used motion direction stimuli, reconstruction strength built up slowly during the stimulus presentation, reaching its maximum only after stimulus offset. Possibly, different stimulus material with faster build-ups might help to track the neural representation during early encoding stages. Another possibility why serial dependence emerged in our study only during the retro-cue epoch is that the neural representation of the stimulus during its post-encoding processing was noisier than during its encoding and thus potentially more susceptible to bias. However, we found comparable time courses of the reconstruction fidelities between S1, S2, and retro-cue epochs. On the other hand, a biased representation, which represents a small and hard-to-detect neural effect, should be easier to observe for less noisy data. The fact that we found a significant neural bias only during the potentially 'noisier' retro-cue epoch makes our finding even more noteworthy.

It is also worth mentioning that the neural attractive bias in our study was about three times larger than the behavioral attraction bias. As both measures provided an identical metric (angle degree), one could expect that their magnitudes should be directly comparable. However, we speculate that these magnitudes inform only about the direction of the bias and their significant difference from zero, thus they operate on different scales and are not directly comparable. In line with this, *Hallenbeck et al., 2021* showed that fMRI-based reconstructed orientation bias and behavioral bias correlated on both individual and group levels, despite strong magnitude differences.

In summary, our study provides a direct neural signature of serial dependence that mirrors the attractive behavioral bias. This demonstrates that serial dependence in working memory directly acts on memory representations and biases them toward the past. The attractive distortion of the current

representation happened after object encoding, suggesting that read-out processes during object decoding are important for the emergence of serial dependence.

# Materials and methods

## Key resources table

| Reagent type (species) or resource | Designation | Source or reference | Identifiers | Additional information |
|---|---|---|---|---|
| Software, algorithm | MATLAB, 2019 | https://www.mathworks.com/ | RRID:SCR_001622 | |
| Software, algorithm | Circular statistics | https://de.mathworks.com/matlabcentral/fileexchange/10676-circular-statistics-toolbox-directional-statistics | RRID:SCR_016651 | |
| Software, algorithm | BADS | *Acerbi and Ma, 2017* | https://arxiv.org/abs/1705.04405 | |
| Software, algorithm | EzyFit v2.44 | https://de.mathworks.com/matlabcentral/fileexchange/10176-ezyfit-2-44 | | |
| Software, algorithm | Python 38 | https://www.anaconda.com/docs/main | | |
| Software, algorithm | Spyder v.4.1.5 | https://www.spyder-ide.org/ | RRID:SCR_017585 | |
| Software, algorithm | autoreject v0.2.2 | https://autoreject.github.io/stable/index.html | RRID:SCR_022515 | |
| Software, algorithm | MNE software v0.22.0 | https://mne.tools/stable/index.html | RRID:SCR_005972 | |
| Software, algorithm | NumPy v1.19.2 | https://numpy.org/ | RRID:SCR_008633 | |
| Software, algorithm | Pandas v1.1.3 | https://pandas.pydata.org/ | RRID:SCR_018214 | |
| Software, algorithm | SciPy v1.5.2 | https://scipy.org/ | RRID:SCR_008058 | |

## Experimental design

### Subjects

Thirteen participants were recruited from the Goethe-University Frankfurt and the Fresenius University of Applied Sciences Frankfurt. All subjects reported normal or corrected-to-normal vision and were screened for red–green blindness. One subject participated as a pilot subject for only one MEG session, one subject was excluded after the first behavioral training session because of dependency on glasses, and one subject dropped out after the first MEG session due to exhaustion because of the MEG setting. This resulted in a sample of 10 subjects (3 male), aged between 22 and 30 years (mean: 25.2 years). All subjects gave informed consent and received either a compensation of €10/hr or course credit. The study was approved by the Ethics Committee of the Medical Faculty of the Goethe-University Frankfurt am Main.

The choice for the sample size was based on a priori power calculation. At the time of the sample size calculation, there were no comparable EEG or MEG studies to inform our power calculation. Thus, we based our calculation on a robust serial dependence effect that we found in a behavioral study including four different experiments with overall more than 100 participants with 1632 trials each (*Fischer et al., 2020*). Based on the contrast between target and non-target with an effect size of 1.359 in Experiment 1, a power analysis with 80% desired power led to a small, estimated sample size of six subjects. However, we expected that the detection of the neural signature of this effect would require more participants. Thus, we based our final power calculation on a much smaller behavioral effect, that is, the modulation of serial dependence by the context-feature congruency. We focused again on the results from Experiment 1 of our previous study that used color as the context feature for retro-cueing (*Fischer et al., 2020*), as we planned to use the same paradigm for the MEG study. This color congruency effect resulted in a more conservative power estimate: Based on an effect size of 0.856 in that experiment, a sample size of $n = 10$ should yield a power of 80% with two MEG sessions per subject.

## Experimental paradigm

### Stimuli

Random dot patterns (RDP) were presented centrally on the screen and consisted of 200 dots colored in red (RGB: 255, 0, 0) or green (RGB: 0, 86.4105, 0) on a black background. The green was adjusted to

match the luminance of the red in DKL color space and then transferred back to RGB space. The dots were presented within an invisible circular aperture which had a radius of 7.5° of visual angle. The dots had a diameter of 0.15° of visual angle and were placed randomly within the circular aperture of the RDP at stimulus onset. The dots moved with a velocity of 3.5°/s and were fully coherent in a direction randomly drawn from a pool of directions between 5° and 355° spaced 10° from one another, therefore avoiding cardinal directions. Dots reaching the edge of the aperture were repositioned randomly on the edge of the opposing side of the aperture, so that dot density was kept constant throughout the presentation. Throughout the whole experiment, a white fixation square with a diagonal of 0.15° of visual angle was presented centrally on the screen, except for the cue presentation, when the fixation square changed its color to red or green to cue which item should be reported. The item was reported by adjusting a randomly oriented line to match the reported direction. The response line was white, with a width of 0.6° and a length equaling the dot field radius. The starting point of the line was the fixation square, and the end point could be altered so that the line could point in all possible directions.

## Procedure

The experiment consisted of a delayed-estimation task. Specifically, each trial began with the presentation of the first stimulus (S1) for 200 ms followed by a noise mask for 100 ms consisting of dots moving with 0% coherence (i.e., randomly) and of the same color as the preceding RDP. After a 1000-ms interval (ISI), the second stimulus (S2) and its noise mask were presented for 200 and 100 ms, respectively (*Figure 1*). Subjects were asked to memorize the motion direction of both RDP for a delay of 1000 ms. Then the fixation square changed its color to red or green for 1500 ms, thereby cueing which motion direction became a target item and had to be reported. Immediately after cue offset, a randomly oriented line was presented. Subjects had to report the target motion direction by rotating the line by moving an MEG-compatible track ball horizontally. No time limit was given for the response, and subjects were encouraged to work as precisely as possible. After adjusting the line direction, subjects had to confirm their response by pressing the left trackball button. At the end of each trial, a fixation screen of 2000–2500 ms (jittered in steps of 10 ms, randomized) was presented. Subjects were instructed to fixate the fixation square throughout the whole experiment.

Between trials, 36 possible direction differences (–170°:10°:180°) were balanced so that every possible direction difference between the items of the current trial and the previous target occurred equally often. This was done with the restriction that the items of the current trial were not allowed to have the same motion direction difference to the previous target item, that is, cued for report in the previous trial. Thereby, it was ensured that the items within a trial could not have the same motion direction. Furthermore, the congruence of color and temporal position between target items of the previous and current trial was balanced, which automatically led to a balancing for the non-target item of the current trial, too, and ensured that in each trial the two items had different colors from one another.

Every subject completed 1022 trials in two sessions on different days. Each session lasted approximately 90 min and was divided into 7 blocks of 73 trials with breaks in between. After the completion of each block, subjects received feedback ('You deviated by more than 30° in XX% of trials.' – German: 'Du bist in XX% der Durchgänge um mehr als 30° abgewichen.') about their performance in the previous block, displayed on the screen. Subjects were seated at a viewing distance of approximately 50 cm from the display. MATLAB software with the Psychophysics Toolbox extensions was used for stimulus generation and presentation. A PROPixx projector (VPixx Technologies Inc) with a resolution of 1920 × 1080 and running with a 120 Hz refresh rate was used.

Prior to the two MEG sessions, each subject was invited for a behavioral practice session. This session consisted of eight blocks of the task and was therefore slightly longer than the MEG experiment. We screened the participants for red–green blindness, checked MEG and MRI contraindications, and instructed them in the behavioral task before participants started to practice the task. In addition to the feedback presented at the end of each block as in the MEG part of the study, subjects received trial-wise feedback in this first session. If their response deviated more than 30° from the cued motion direction, we presented feedback ('>30°!'). Thereby they could practice to be as precise as possible for the MEG sessions.

## MEG recording and preprocessing

### Data acquisition

We recorded MEG with a whole-head MEG system (Omega 2005; VSM MedTech Ltd, Port Coquitlam, Canada) with 275 axial gradiometers (271 active) at a sampling rate of 1200 Hz and without online filtering. Additionally, we recorded electrocardiogram (ECG), vertical and horizontal electrooculograms (EOG) as well as continuous gaze position and pupil dilation of the right eye by an MEG-compatible eye tracker (Eyelink CL 1000, SR Research Ltd). Head position was continuously recorded via head localization coils placed at the nasion and above both ear canal entrances using ear plugs. The initial head position was saved at the beginning of the first MEG session and presented to the participants for repositioning at the beginning of the second session as well as throughout each session if repositioning was necessary. Behavioral responses were recorded with an MEG-compatible trackball (Current Designs Inc), enabling subjects to continuously recall motion directions.

### Preprocessing

MEG data were preprocessed using the python-based M/EEG analysis toolbox MNE (version 0.22.0; *Gramfort et al., 2013*). The MEG signal was first notch filtered at 50 Hz and up to 250 Hz in steps of 50 Hz to remove line noise and then cut into trials from 1000 ms before S1 onset until 500 ms after response onset. Trials containing SQUID jump artifacts were manually identified and excluded. The remaining trials were then high-pass filtered (1 Hz) and decomposed via independent component analysis (ICA) with the FastICA method implemented in MNE. The decomposition was then applied to the notch-filtered continuous data, and ICs representing cardiac and ocular activity were identified via correlation with ECG and EOG signals. The notch-filtered continuous data were then low-pass filtered at 25 Hz and cut for the respective analysis. To analyze reactivations of the previous target direction, the data were cut into one long epoch (last second of the ITI up until the end of the retro-cue delay). For the analysis regarding the directions of the current trial, we used three epochs per trial: S1 encoding and maintenance (called S1 epoch), S2 encoding and maintenance (called S2 epoch), and retro-cue presentation and delay (called retro-cue epoch). The epochs started 200 ms before stimulus/retro-cue onset and lasted until 1500 ms after stimulus/retro-cue onset. For the S1/S2 epochs, only 1300 ms after stimulus onset were included in the analysis as those reflected the presentation + delay time. Those epochs were baseline-corrected (–200 to 0 ms), the previously identified ICs reflecting cardiac and ocular activity were removed, and the data resampled at 300 Hz.

## Statistical analysis

### Behavioral analysis

First, we excluded trials in which the response error was at least 3 SDs higher than the subject's mean response error, or in which the response time exceeded 20 s, indicating potential attentional lapses. We also excluded the first trial of each block, as serial dependence cannot occur on those trials, and demeaned the response errors by subtracting the overall mean response error of a participant from each individual response error to remove general individual response biases independent of serial dependence.

The evaluation of serial dependence was based on individual response errors, defined as the deviation between presented and entered direction. The errors were sorted regarding the difference between the target stimulus of the current trial and the target stimulus from the previous trial, as well as the relation of difference between the current item and the item of the previous trial (CW or counter-clockwise [CCW]). The difference was computed by subtracting the direction of the current item from the direction of the item of the previous trial. Therefore, when the current item was oriented more CW or more CCW, this resulted in a negatively or positively signed distance, respectively. A mean response error for a signed distance (distance * relation) deviating from 0 indicated a systematic response bias. When the sign of this systematic bias matched the sign of the distance between the directions, it indicated an attractive response bias. Conversely, an opposite sign of the systematic bias compared to the signed distance indicated a repulsive response bias. In addition, the signed response errors were sorted according to the color congruence between the two items (same vs. different color). The individual mean response biases were used to evaluate the serial dependence per color congruence level. We fitted the first derivative of a Gaussian curve DoG; for example, (1), a model which is usually used to describe serial dependence. The DoG, given by

$$y = xawce^{-(wx)^2} \tag{1}$$

was fitted to the pooled mean response biases of all subjects (similar to the procedure by *Fritsche et al., 2017*) per factor level, that is, one data point per subject and distance for the respective factor level. In the DoG, $x$ is the relative direction difference of two stimuli, $a$ is the amplitude of the curve peak, $w$ scales the curve width, and $c$ is the constant $y\sqrt{2}/e^{-0.5}$. The $w$ parameter was constrained to a value range of 0.01–0.1. We optimized the log likelihoods of our curve fitting using Bayesian adaptive direct search (BADS; *Acerbi and Ma, 2017*). BADS alternates between a series of fast, local Bayesian optimization steps and a systematic, slower exploration of a mesh grid. To estimate the variability of the parameters $a$ and $w$, we bootstrapped the DoG curve fit 1000 times, sampling the data with replacement on each iteration, and computed the standard deviation of the resulting bootstrapped distributions of $a$ (see *Fischer et al., 2020*) for a similar procedure. To assess the significance of serial dependence on a group level, we used permutation tests, that is, we randomly inverted the signs of each participant's mean response error, fitted a new DoG model to the pooled group data, and collected the resulting amplitude parameters $a$ in a permutation distribution. We repeated this permutation procedure 1000 times and reported the percentage of permutations that led to equal or higher values for $a$ than the one estimated for the empirical data as p-values. The significance level was set to $\alpha$ = 0.05 (one-sided permutation test).

## MEG analysis
### IEM analysis
To reconstruct motion directions from neural signals, we used the so-called IEM technique (*Sprague et al., 2018*). We built an encoding model characterizing motion direction-selective responses in the MEG signal with the underlying assumption that the MEG signal in each sensor reflects a weighted sum of neural activity revealed by different motion directions. Our model consisted of a basis set of 18 channels spanning the room of 0–360° with centers in steps of 20°. The channels had the shape of sinusoids raised to the 18th power (adapted from *Ester et al., 2015*; *Sprague, 2016*). This model was applied to the MEG data, which was separated into a training and a test dataset. For each subject, we trained the model on one MEG session and tested it on the other one in two iterations, so that each MEG session was once training and once test data, and training and test data were independent. To analyze the currently relevant motion directions, we applied this reconstruction approach to the three types of epochs (S1 epoch, S2 epoch, and post-cue epoch) separately, and in each epoch, the currently relevant direction was the aim of the reconstruction (S1 epoch: S1 direction, S2 epoch: S2 direction, post-cue epoch: cued target direction). To increase the signal-to-noise ratio, the epoched data were averaged across time bins of 100 ms, centered on 50 ms, 150 ms, etc. after epoch onset. This led to 13 analysis time points for the S1 and S2 epoch and 15 analysis time points for the post-cue epoch. For each epoch, the directions were reconstructed for each time point separately in a time point by time point fashion. To check for reactivations of the previous target direction, we applied the same analysis on the last second of the ITI and the whole time course of the current trial, resulting in 51 analysis time points. This time, the aim of the reconstruction was the previous target direction; therefore, the data were trained and tested on this direction.

In the first step, we modeled the activation of each sensor as a weighted sum of filter responses:

$$B_1 = C_1 W \tag{2}$$

$B_1$ is the MEG signal in each sensor at a given time point ($n$ trials * $n$ sensors), $C_1$ the modeled response of each channel given the presented direction in each trial ($n$ trials * $n$ channels) and $W$ a weight matrix representing the contribution of each channel to the measured signal in each sensor ($n$ channels * $n$ sensors). $W$ was estimated by using ordinary least-squares regression:

$$\hat{W} = \left(C_1^T C_1\right)^{-1} C_1^T B_1 \tag{3}$$

In the second step, we inverted $W$ and applied it to the independent test MEG data $B_2$ ($n$ sensors * $n$ trials) to compute the estimated channel responses $\hat{C}_2$ ($n$ channels * $n$ trials):

$$\hat{C}_2 = \left(\hat{W}^T \hat{W}\right)^{-1} \hat{W}^T B_2 \tag{4}$$

The estimated channel responses $\hat{C}_2$ were transformed from the channel space back to the direction space by weighting $\hat{C}_2$ with each channel's sensitivity from the basis set. Those reconstructed channel responses were then circularly shifted to a common center (180°). This whole reconstruction procedure was done separately per subject, session, epoch, and time point, and this was iterated 20 times. Across those iterations, the centers of the basis set were shifted by 1° so that the channel centers were evenly spaced in steps of 1° across the whole direction space as a result of those iterations. The single-trial reconstructions were then analyzed in two different ways, described in the following paragraphs.

## Shifts of the reconstructed direction

The main goal of our analysis was to examine potential shifts of the neural representation in relation to the target item of the previous trial. Therefore, the single-trial reconstructions were ordered in two groups, those with a previous target that was CW oriented in relation to the currently relevant item and those with a previous target that was CCW oriented. The CCW reconstructions were flipped along the direction space. Thereby, for both CW and CCW trials, a negative deviation of the maximum of the reconstruction from 180° indicated an attraction toward the previous target, whereas a positive deviation indicated a repulsion. Those reconstructions were then first averaged within each possible motion direction and then across them to account for different presentation numbers of the directions. This averaged reconstruction was obtained for each iteration (see above) and then later averaged across sessions. Thereby, the analysis resulted in a reconstruction per participant, epoch, and time point. In the next step, the fidelity of those reconstructions was assessed per epoch and time point to identify time points with significant direction information in the MEG signal. A fidelity above 0 indicates a meaningful direction signal and was obtained by projecting the polar vector of the reconstruction at every direction angle (in steps of 1°) onto the common center (180°). The length of this projected vector indicates the information strengths of the reconstruction at each angle. The fidelity $F$ was obtained by averaging those information strengths:

$$F = mean\left(r\left(\theta\right)\cos\left(\theta\right)\right) \tag{5}$$

We identified the time points in each epoch with significant direction reconstruction fidelity by computing a one-sided cluster-based permutation test (as implemented in MNE) against zero. As the effect of the neural shift was expected to be very small, we averaged the reconstructions in each epoch only over those time points that showed significant reconstruction fidelity and thus interpretable data. To examine systematic shifts, we then tested if the maximum of the reconstruction was systematically different from the common center (180°). For display purposes, we subtracted the reconstructed maximum from 180° to compute the direction shifts. A positive shift thus reflected attraction, and a negative shift reflected repulsion.

## Statistical inference of shifts

To assess the significance of potential shifts, we used a bootstrapping procedure similar to *Ester et al., 2020*. To this aim, we randomly selected reconstructions from 10 participants (with replacement), then computed the mean reconstruction over the resampled participants and collected the shift. This was done on 10,000 iterations. The distribution of shifts was then used to examine if the observed shift was significantly larger than zero. A shift was significant when 95% of the resampling iterations led to a shift equal to or higher than 0°, that is, we conducted a one-sided bootstrapping test as we expected an attraction, which was indicated by a shift larger than 0°. To obtain an indicator for the variability of the shifts, the middle 95% of shifts resulting from the described resampling procedure are reported. Thus, our bootstrapping procedure relied on (1) detecting an offset against zero and (2) evaluating the robustness of the observed effect across participants. As such, it contrasts with the frequently used permutation approach that assesses whether an empirical neural shift is more extreme than the permutation distribution. The permutation approach seemed more suited to assess the magnitude of the shift, which in our study was not a priority. Therefore, we reasoned that bootstrapping was better suited to assess the direction of the neural shift and its robustness across participants.

## Correlation between neural representation and behavioral response

In addition, we correlated the maximum of single-trial reconstructions with the upcoming behavioral response to test whether shifts in behavior and in the neural signal on a single-trial level were systematically related to each other. Both behavioral and neural shifts in any direction (CW or CCW) in a single-trial response can originate from or be modulated by different sources like, for example, noise in the motor response, but they can also result from attractive serial dependence that can be observed by averaging over trials. We collected the reconstructions before flipping them with regard to the previous target and removed trials where the behavioral response deviated more than three SDs from the individual response error. Then we computed the circular correlation between the deviation of the reconstructed maximum from the presented direction and the response error for each participant, session, epoch, and time point. For S1 and S2 epochs, only epochs from trials where the respective item was cued as the response target were included.

## Statistical inference of correlations

The obtained correlation coefficients were averaged across sessions and then time points for each epoch with a correlation significantly differing from zero were identified by computing a one-sided cluster-based permutation test (as implemented in MNE) against zero.

## Reconstruction of previous target direction

We also examined whether the neural signals of the current trial contained memory traces of the previous target direction. Therefore, another IEM analysis was employed, but this time trained and tested on the target direction of the previous trial throughout the last second of the ITI and the current trial. Time points that contained significant direction information from the previous target were identified by computing the fidelity of the reconstructions as described above and tested against zero with a one-sided cluster-based permutation test.

## Multiple comparison correction

In order to correct for multiple comparisons, we applied an FDR correction (Benjamini–Hochberg) to those analyses that resulted in one p-value per epoch, that is, the bootstrapping tests of the shifts against zero, and reported these corrected values alongside the original ones. Furthermore, for cluster-based permutation tests that were applied to each epoch individually, we multiplied p-values higher than the minimally possible ($p < 0.002$) for significant clusters by the number of epochs (3), thus resulting in a Bonferroni correction.

## Software

All behavioral analysis was performed with *MATLAB, 2019* and the following toolboxes/functions: Circular Statistics Toolbox (*Berens, 2009*), BADS (*Acerbi and Ma, 2017*), and EzyFit v2.44 (*Moisy, 2016*). The MEG analysis was performed with Python 3.8 in the distribution v.2020.11 (*Anaconda, 2020*) using spyder v.4.1.5 (*Raybaut, 2009*) and the following toolboxes: autoreject v0.2.2 (*Jas et al., 2017*), MNE v0.22.0 (*Gramfort et al., 2013*), numpy v1.19.2 (*Harris et al., 2020*), pandas v1.1.3 (*McKinney, 2010*; *The Pandas Development Team, 2020*), and scipy v1.5.2 (*Virtanen et al., 2020*).

## Acknowledgements

We thank Magdalena Feldmann, Nicole Huizinga, Philipp Deutsch, and Susanna Wenzel for their help in data collection and Benjamin Rahm for helpful discussions.

# Additional information

### Funding

| Funder | Grant reference number | Author |
|---|---|---|
| German Academic Scholarship Foundation | | Cora Fischer |

The funders had no role in study design, data collection, and interpretation, or the decision to submit the work for publication.

### Author contributions
Cora Fischer, Conceptualization, Formal analysis, Investigation, Methodology, Visualization, Writing – original draft, Writing – review and editing; Jochen Kaiser, Conceptualization, Methodology, Supervision, Writing – review and editing; Christoph Bledowski, Conceptualization, Methodology, Supervision, Visualization, Writing – original draft, Writing – review and editing

### Author ORCIDs
Cora Fischer ⓘ https://orcid.org/0000-0002-5668-9439
Jochen Kaiser ⓘ https://orcid.org/0000-0001-6740-5000
Christoph Bledowski ⓘ https://orcid.org/0000-0003-3628-7683

### Ethics
All subjects gave informed consent and received either a compensation of €10/hr or course credit. The study was approved by the Ethics Committee of the Medical Faculty of the Goethe-University Frankfurt am Main.

Reviewer #1 (Public review): https://doi.org/10.7554/eLife.99478.3.sa1
Reviewer #2 (Public review): https://doi.org/10.7554/eLife.99478.3.sa2
Reviewer #2 (Public review): https://doi.org/10.7554/eLife.99478.3.sa3
Author response https://doi.org/10.7554/eLife.99478.3.sa4

# Additional files

### Supplementary files
MDAR checklist

### Data availability
Code, behavioral data, and preprocessed MEG data used for the present analyses are available on OSF via https://osf.io/yjc93/. Due to storage limitations (50 GB), only the preprocessed MEG data for the main IEM analyses focusing on the current direction were uploaded. Researchers interested in accessing the raw MEG data (ca. 220GB) should contact the principal investigator Christoph Bledowski (bledowski@em.uni-frankfurt.de), providing a rationale for their non-commercial request. Upon review, they will be granted access to the raw data.

The following dataset was generated:

| Author(s) | Year | Dataset title | Dataset URL | Database and Identifier |
|---|---|---|---|---|
| Fischer C, Kaiser J, Bledowski C | 2024 | A direct neural signature of serial dependence | https://osf.io/yjc93/ | Open Science Framework, yjc93 |

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

# Appendix 1

## Correlation between the neural shift and individual serial dependence parameters

In addition to the single-trial-based correlation between neural and behavioral shifts reported in the main manuscript, we calculated correlations between the neural shift and individual *a* and *w* parameters obtained from DoG serial dependence fits on a between-subject level. This analysis revealed no significant correlations during the S1 and S2 epochs for neither *a* nor *w* parameter (*a* parameter, S1: $r = 0.01$, p = 0.9767; S2: $r = 0.08$, p = 0.8114; *w* parameter: S1: $r = 0.36$, p = 0.3096; S2: $r = 0.30$, p = 0.4054). While during the retro-cue epoch, there was no significant correlation with the *a* parameter either ($r = -0.35$, p = 0.3258), we did observe a significant correlation between individual neural shifts and *w* parameters ($r = -0.70$, p = 0.0246). This means that smaller individual *w* parameters indicating a broader tuning of serial dependence went along with larger neural shifts.

This further demonstrates that the neural shift during the retro-cue epoch reflected behavioral serial dependence. These results are displayed in *Appendix 1—figure 1*.

It is important to note that for the calculation of the neural shift, all trials entered the analysis to increase the signal-to-noise ratio, that is, it included many trials where current and previous targets were separated by, for example, 100° or more. These trials were unlikely to produce serial dependence. Subjects with a more broadly tuned serial dependence had more inter-item differences that showed a behavioral attraction and therefore more trials affected by serial dependence that entered the calculation of the neural shift. In contrast, individual differences in the amplitude (*a*) parameter were most likely too small, and higher individual amplitude did not involve more trials as compared to smaller amplitude to affect the neural bias in a way to be observed in a significant correlation.

## Different codes for reactivation of previous information and reconstructed biases of current information

In the retro-cue epoch, we observed both an attractive bias toward the target of the previous trial and a transient reactivation of the previous target itself. Therefore, one alternative explanation of the observed attractive bias could be a reactivation of the previous target in the same neural code as the current target that in sum led to an attractive shift of the reconstruction. To rule out this explanation, we ran a cross-validation analysis. Therefore, we trained the encoding model on the currently relevant item (S1 during the S1 epoch, S2 during the S2 epoch, and the cued item during the retro-cue epoch) and aimed to reconstruct the motion direction of the previous trial.

This analysis revealed that during the retro-cue epoch, where we observed the neural shift toward the past, as well as during the S1 epoch, there was no reactivation in the same neural code as the current representation. In contrast, a reactivation of target information from the previous trial using the same neural code could be observed during stimulus presentation in the S2 epoch. This indicated that during the S2 phase, information from the current S2 and the previous target was reconstructed in the same neural code. These results are displayed in *Appendix 1—figure 2*.

## Eye movement control

Participants were instructed to maintain fixation on a small square at the center of the screen throughout each trial, and independent component analysis (ICA) was used during preprocessing to remove any eye movement artifacts. However, recent studies showed that systematic eye movements can potentially confound decoding results (e.g., *Mostert et al., 2018*; *Quax et al., 2019*). In the present study, participants viewed dynamic stimuli (moving dot patterns) and had to remember a spatial feature (motion direction), which made systematic eye movement behavior more likely. Therefore, we thoroughly tested (1) whether eye movements were associated with the presented/remembered motion direction and if so, (2) whether eye movements could have driven our MEG results.

As a first step of our eye movement control analysis, we tested how much of the variance of gaze direction could be explained by the stimulus direction that was either currently presented or had to be maintained at a given moment in time. More specifically, we first transformed the *x* and *y* positions measured by the eye tracker (in relation to a baseline from 200 ms until trial onset) into polar coordinates to obtain a gaze direction in degrees. This transformation was performed

on the same time points as the MEG analysis, that is, gaze directions were averaged within bins of 100 ms. Then we computed the circular variance of these gaze directions separately for each specific stimulus direction and time bin per subject. This circular variance was compared to a random distribution of gaze direction variance. To obtain this distribution, we shuffled the motion direction labels randomly over trials and then computed the circular variance for each pseudo direction, which was unrelated to the direction that was actually presented in the trial. The empirical and shuffled variances were averaged across directions, and then a ratio was computed between them. The resulting value reflects the extent to which the variance of the stimulus-related gaze directions was reduced compared with random gaze direction patterns. This procedure was able to reveal not only systematic gaze directions that were in accordance with the stimulus direction or the opposite direction, but also picked up all stimulus-related gaze directions, even if the relation differed across participants or time. We identified time points with a significant amount of stimulus-related gaze directions in each analyzed epoch with cluster-based permutation tests. *Appendix 1—figure 3a* shows that gaze directions were systematically related to the presented or remembered motion direction at different time points during each epoch. However, stimulus-related gaze directions differed considerably between participants as indicated by the standard errors.

In the second step, we plotted the time courses of stimulus-related gaze direction separately for each subject. This is shown in *Appendix 1—figure 3b*. This revealed that while some participants showed a high amount of stimulus-related gaze directions, others showed very little or no stimulus-related gaze directions at all. In *Appendix 1—figure 3a*, we marked the time courses in dark red for subjects with the highest amount of direction-related gaze positions (high) and dark green for those with the lowest amount (low), with gradients in between.

In the third step, we addressed the question of how the stimulus-related gaze directions related to our MEG results. To this aim, we plotted the MEG-based reconstruction fidelity of presented and remembered motion direction separately for each subject and epoch and, crucially, coded them by the amount of stimulus-related gaze directions shown by each subject. This is shown in *Appendix 1—figure 3c*. Comparing the stimulus reconstruction fidelity time courses with the individual color codes derived from the gaze direction analyses revealed that the reconstruction fidelities of each participant did not correspond to their amount of stimulus-related gaze behavior (compare *Appendix 1—figure 3b* with *Appendix 1—figure 3c*). This suggests that there was no systematic relationship at any time point between the fidelity measure and the amount of direction-related gaze positions.

To support this observation statistically, we calculated Spearman rank correlations between the amount of stimulus-related gaze directions and reconstruction fidelities individually for each time point of each epoch. The resulting p-values are shown in *Appendix 1—figure 3d*. Even before correcting for multiple comparisons, the p-values were consistently above 0.05, except for one single time point in the retro-cue epoch. Moreover, the p-values within each epoch did not show any consistent time profile but varied considerably between time points, supporting our interpretation that MEG-based reconstruction fidelity was not driven by the amount of stimulus-related gaze directions.

In consequence, we also did not find any relationship between the attractive neural shift that was based on these reconstructed MEG signals (i.e., the main finding of our study) and stimulus-related gaze directions. Specifically, there was no correlation between each subject's rank of the strength of neural shift toward the previous target and the subject's rank of the amount of stimulus-related gaze directions in any of the epochs (S1: p = 0.986, S2: p = 0.260, retro-cue: p = 0.244).

We conclude that our MEG results were not crucially driven by eye movements.

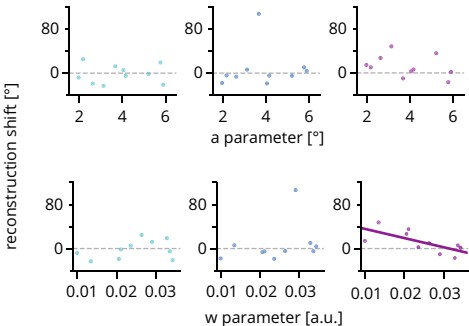

**Appendix 1—figure 1.** Correlation between the neural shift and individual serial dependence parameters. Relation between the individual *a* (upper panels) and *w* (lower panels) parameters of a derivative-of-Gaussian (DoG) serial dependence fit (*x*-axis) and the neural reconstruction shift (*y*-axis). Colored circles show individual values for the S1 (left panels), S2 (middle panels), and retro-cue epochs (right panels). Solid lines depict significant correlations, dashed gray lines mark a reconstruction shift of 0°. *N* = 10.

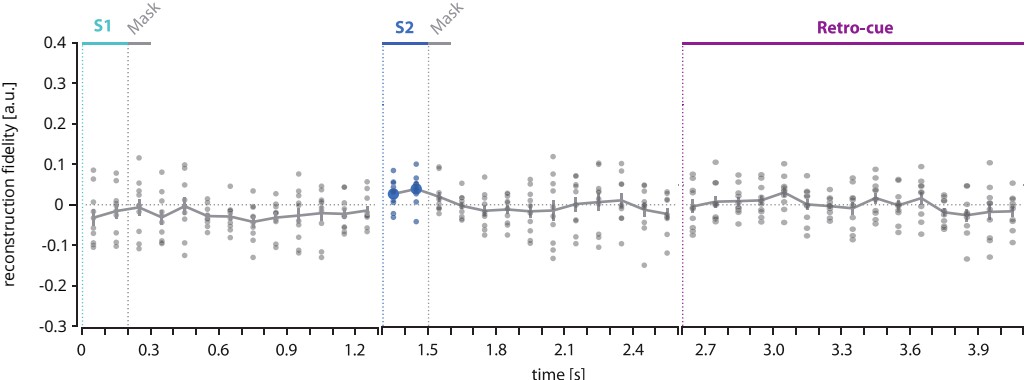

**Appendix 1—figure 2.** Cross-validated reconstruction fidelity of previous direction, model trained on currently relevant direction. Fidelity of reconstructed direction of the previous target using a model that was trained on the currently relevant direction. The left panel shows the fidelity of the previous target during the S1 epoch, the middle panel depicts the fidelity during the S2 epoch, and the right panel shows the fidelity of the cued direction (either S1 or S2). Small circles show the fidelity of each participant. Colored circles indicate time points with a fidelity significantly greater than zero. Cluster-based permutation test within each epoch. *N* = 10.



**Appendix 1—figure 3.** Systematic eye movements and reconstruction time courses were uncorrelated. (**a**) Amount of stimulus-related gaze direction (indicated by variance reduction) during each epoch as a function of time with error bars indicating standard errors and colored circles indicating time points with a variance reduction significantly greater than zero. Cluster-based permutation test within each epoch; p < 0.05. N = 10. (**b**) Time courses of individual stimulus-related gaze directions. Dark red indicates the subject with the highest average variance reduction in a given epoch, while green indicates the one with the smallest average variance reduction, with color gradients in between according to the individual rank of average variance reduction. (**c**) Time courses

*Appendix 1—figure 3 continued*
of individual reconstruction fidelities. Lines are colored as in (b), reflecting the stimulus-related gaze direction rank of a subject in a given epoch. (**d**) Time courses of p-values (uncorrected for multiple comparisons) of time point-by-time point Spearman rank correlations between the systematic gaze directions as shown in (**b**) and the reconstruction fidelity as shown in (**c**). $N = 10$.

