## [Editor Report · eLife Assessment]

This study reveals a neural signature of a common behavioural phenomenon: serial dependence, whereby estimates of a visual feature (here motion direction) are attracted towards the recent history of encoded and reported stimuli. The study provides **solid** evidence that this phenomenon arises primarily during working memory maintenance. The pervasiveness of serial dependencies across modalities and species makes these findings **important** for researchers interested in perceptual decision-making across subfields.

---

## [Referee Report · Reviewer #1 (Public review)]

This study uses MEG to test for a neural signature of the trial history effect known as 'serial dependence.' This is a behavioral phenomenon whereby stimuli are judged to be more similar than they really are, in feature space, to stimuli that were relevant in the recent past (i.e., the preceding trials). This attractive bias is prevalent across stimulus classes and modalities, but a neural source has been elusive. This topic has generated great interest in recent years, and I believe this study makes a unique contribution to the field.

Specifically, while previous neuroimaging studies have found apparent reactivations of previous information, or repulsive biases that may indirectly relate to serial dependence, here Fischer at al. find an attractive bias in neural activity patterns that aligns with the direction of the behavioral effect. Moreover, the data show that the bias emerges later in a trial, after perceptual encoding, which speaks to an ongoing debate about whether such biases are perceptual or decisional.

The revised preprint thoroughly addresses many of the initial concerns, but the results are still open to interpretation. For instance, the model training/testing regime allows that some training data timepoints may be inherently noisier than others (e.g., delay period more so than encoding), and potentially more (or differently) susceptible to bias. The S1 and S2 epochs show no attractive bias, but they may also be based on more high fidelity training sets (i.e., encoding), and therefore less susceptible to the bias that is evident in the retrocue epoch. So, the results could reflect that serial dependence is indeed a post-perceptual process, or it may instead be that the WM representations, as detected with these MEG analyses, become noisier and more subject to reveal the attractive bias over time.

The results are intriguing, but the study was not powered to examine whether there is any feature-specificity to the neural bias (e.g., whether it matches the behavioral pattern that biases are amplified within a particular range of feature distances between stimuli). Nor do analyses get at temporally precise information about when attractive and repulsive biases appear, which would help to better reconcile the work with previous findings. As in, the reconstructions average across coarse trial epochs. The S1 and S2 reconstructions show no attractive bias, and appear to show subtle repulsion, but if the timing were examined more precisely, we might see repulsion magnified at earlier timepoints that shift toward attraction at later time points, thereby counteracting the effect. That is to say that the averaging approach, across feature values and timepoints, still leaves these important theoretical questions unresolved.

Nonetheless, the work marks an important step in identifying the neurophysiological bases of serial dependence. Ideally, all of the data, including the eye-tracking, would be made available so that others might try to address some of these follow-up questions.

---

## [Referee Report · Reviewer #2 (Public review)]

Summary:

The study aims to probe the neural correlates of visual serial dependence - the phenomenon that estimates of a visual feature (here motion direction) are attracted towards the recent history of encoded and reported stimuli. The authors utilize an established retro-cue working memory task together with magnetoencephalography, which allows to probe neural representations of motion direction during encoding and retrieval (retro-cue) periods of each trial. The main finding is that neural representations of motion direction are not systematically biased during the encoding of motion stimuli, but are attracted towards the motion direction of the previous trial's target during the retrieval (retro-cue period), just prior to the behavioral response. By demonstrating a neural signature of attractive biases in working memory representations, which align with attractive behavioral biases, this study highlights the importance of post-encoding memory processes in visual serial dependence.

Strengths:

The main strength of the study is its elegant use of a retro-cue working memory task together with high temporal resolution MEG, enabling to probe neural representations related to stimulus encoding and working memory. The behavioral task elicits robust behavioral serial dependence and replicates previous behavioral findings by the same research group. The careful neural decoding analysis benefits from a large number of trials per participant, considering the slow-paced nature of the working memory paradigm. This is crucial in a paradigm with considerable trial-by-trial behavioral variability (serial dependence biases are typically small, relative to the overall variability in response errors). While the current study is broadly consistent with previous studies showing that attractive biases in neural responses are absent during stimulus encoding (prev. studies reported repulsive biases), to my knowledge, it is the first study showing attractive biases in current stimulus representations during working memory. The study also connects to previous literature showing reactivations of previous stimulus representations, although the link between reactivations and biases remains somewhat vague in the current manuscript. Together, the study reveals an interesting avenue for future studies investigating the neural basis of visual serial dependence.

Weaknesses:

The main weakness of the current manuscript is that the authors could have done more analyses to address the concern that their neural decoding results are driven by signals related to eye movements. The authors show that participants' gaze position systematically depended on the current stimuli's motion directions, which, together with previous studies on eye movement-related confounds in neural decoding, justifies such a concern. The authors seek to rule out this confound by showing that the consistency of stimulus-dependent gaze position does not correlate with (a) the neural reconstruction fidelity and (b) the attractive shift in reconstructed motion direction. However, the authors' approach of quantifying stimulus-dependent eye movements only considers gaze angle and not gaze amplitude, and thus potentially misses important features of eye movements that could manifest in the MEG data. Moreover, it is unclear whether the gaze consistency metric should correlate with attractive history biases in neural decoding, if there were a confound. These two concerns could be potentially addressed by (1) directly decoding stimulus motion direction from x-y gaze coordinates and relating this decoding performance to neural reconstruction fidelity, and (2) investigating whether gaze coordinates themselves are history-dependent and are attracted to the average gaze position associated with the previous trials' target stimulus. If the authors could show that (2) is not the case, I would be much more convinced that their main finding is not driven by eye movement confounds.

The sample size (n = 10) is definitely at the lower end of sample sizes in this field. The authors collected two sessions per participant, which partly alleviates the concern. However, given that serial dependencies can be very variable across participants, I believe that future studies should aim for larger sample sizes.

It would have been great to see an analysis in source space. As the authors mention in their introduction, different brain areas, such as PPC, mPFC and dlPFC have been implicated in serial biases. This begs the question which brain areas contribute to the serial dependencies observed in the current study? For instance, it would be interesting to see whether attractive shifts in current representations and pre-stimulus reactivations of previous stimuli are evident in the same or different brain areas.

---

## [Referee Report · Reviewer #2 (Public review)]

Summary:

The study aims to probe the neural correlates of visual serial dependence - the phenomenon that estimates of a visual feature (here motion direction) are attracted towards the recent history of encoded and reported stimuli. The authors utilize an established retro-cue working memory task together with magnetoencephalography, which allows to probe neural representations of motion direction during encoding and retrieval (retro-cue) periods of each trial. The main finding is that neural representations of motion direction are not systematically biased during the encoding of motion stimuli, but are attracted towards the motion direction of the previous trial's target during the retrieval (retro-cue period), just prior to the behavioral response. By demonstrating a neural signature of attractive biases in working memory representations, which align with attractive behavioral biases, this study highlights the importance of post-encoding memory processes in visual serial dependence.

Strengths:

The main strength of the study is its elegant use of a retro-cue working memory task together with high temporal resolution MEG, enabling to probe neural representations related to stimulus encoding and working memory. The behavioral task elicits robust behavioral serial dependence and replicates previous behavioral findings by the same research group. The careful neural decoding analysis benefits from a large number of trials per participant, considering the slow-paced nature of the working memory paradigm. This is crucial in a paradigm with considerable trial-by-trial behavioral variability (serial dependence biases are typically small, relative to the overall variability in response errors). While the current study is broadly consistent with previous studies showing that attractive biases in neural responses are absent during stimulus encoding (prev. studies reported repulsive biases), to my knowledge, it is the first study showing attractive biases in current stimulus representations during working memory. The study also connects to previous literature showing reactivations of previous stimulus representations, although the link between reactivations and biases remains somewhat vague in the current manuscript. Together, the study reveals an interesting avenue for future studies investigating the neural basis of visual serial dependence.

Weaknesses:

The main weakness of the current manuscript is that the authors could have done more analyses to address the concern that their neural decoding results are driven by signals related to eye movements. The authors show that participants' gaze position systematically depended on the current stimuli's motion directions, which, together with previous studies on eye movement-related confounds in neural decoding, justifies such a concern. The authors seek to rule out this confound by showing that the consistency of stimulus-dependent gaze position does not correlate with (a) the neural reconstruction fidelity and (b) the attractive shift in reconstructed motion direction. However, the authors' approach of quantifying stimulus-dependent eye movements only considers gaze angle and not gaze amplitude, and thus potentially misses important features of eye movements that could manifest in the MEG data. Moreover, it is unclear whether the gaze consistency metric should correlate with attractive history biases in neural decoding, if there were a confound. These two concerns could be potentially addressed by (1) directly decoding stimulus motion direction from x-y gaze coordinates and relating this decoding performance to neural reconstruction fidelity, and (2) investigating whether gaze coordinates themselves are history-dependent and are attracted to the average gaze position associated with the previous trials' target stimulus. If the authors could show that (2) is not the case, I would be much more convinced that their main finding is not driven by eye movement confounds.

The sample size (n = 10) is definitely at the lower end of sample sizes in this field. The authors collected two sessions per participant, which partly alleviates the concern. However, given that serial dependencies can be very variable across participants, I believe that future studies should aim for larger sample sizes.

It would have been great to see an analysis in source space. As the authors mention in their introduction, different brain areas, such as PPC, mPFC and dlPFC have been implicated in serial biases. This begs the question which brain areas contribute to the serial dependencies observed in the current study? For instance, it would be interesting to see whether attractive shifts in current representations and pre-stimulus reactivations of previous stimuli are evident in the same or different brain areas.

---

## [Author Response]

The following is the authors’ response to the original reviews

We were delighted by the reviewers' general comments. We thank the reviewers for their thoughtful reviews, constructive criticism, and analysis suggestions. We have carefully addressed each of their points during the revision of the manuscript.

Unfortunately, after the paper was submitted to eLife, the first author, who ran all the analyses, left academia. We now realized that we currently do not have sufficient resources to perform all additional analyses as requested by the reviewers.3

**Public Reviews:**

**Reviewer #1 (Public Review):**
This study uses MEG to test for a neural signature of the trial history effect known as 'serial dependence.' This is a behavioral phenomenon whereby stimuli are judged to be more similar than they really are, in feature space, to stimuli that were relevant in the recent past (i.e., the preceding trials). This attractive bias is prevalent across stimulus classes and modalities, but a neural source has been elusive. This topic has generated great interest in recent years, and I believe this study makes a unique contribution to the field. The paper is overall clear and compelling, and makes effective use of data visualizations to illustrate the findings. Below, I list several points where I believe further detail would be important to interpreting the results. I also make suggestions for additional analyses that I believe would enrich understanding but are inessential to the main conclusions.(1) In the introduction, I think the study motivation could be strengthened, to clarify the importance of identifying a neural signature here. It is clear that previous studies have focused mainly on behavior, and that the handful of neuroscience investigations have found only indirect signatures. But what would the type of signature being sought here tell us? How would it advance understanding of the underlying processes, the function of serial dependence, or the theoretical debates around the phenomenon?

Thank you for pointing this out. Our MEG study was designed to address two questions: 1. we asked whether we could observe a direct neural signature of serial dependence, and 2. if so, whether this signature occurs at the encoding or post-encoding stage of stimulus processing in working memory. This second question directly concerns the current theoretical debate on serial dependence.

Previous studies have found only indirect signatures of serial dependence such as reactivations of information from the previous trial or signatures of a repulsive bias, which were in contrast to the attractive bias in behavior. Thus, it remained unclear whether an attractive neural bias can be observed as a direct reflection of the behavioral bias. Moreover, previous studies observed the neuronal repulsion during early visual processes, leading to the proposal that neural signals become attracted only during later, post-encoding processes. However, these later processing stages were not directly accessible in previous studies. To address these two questions, we combined MEG recordings with an experimental paradigm with two items and a retro-cue. This design allowed to record neural signals during separable encoding and post-encoding task phases and so to pinpoint the task phase at which a direct neural signature of serial dependence occurred that mirrored the behavioral effect.

We have slightly modified the Introduction to strengthen the study motivation.

(1a) As one specific point of clarification, on p. 5, lines 91-92, a previous study (St. JohnSaaltink et al.) is described as part of the current study motivation, stating that "as the current and previous orientations were either identical or orthogonal to each other, it remained unclear whether this neural bias reflected an attraction or repulsion in relation to the past." I think this statement could be more explicit as to why/how these previous findings are ambiguous. The St. John-Saaltink study stands as one of very few that may be considered to show evidence of an early attractive effect in neural activity, so it would help to clarify what sort of advance the current study represents beyond that.

Thank you for this comment. In the study by St. John-Saaltink et al. (2016), two gratings oriented at 45° and 135° were always presented to either the left or right side of a central fixation point in a trial (90° orientation difference). As only the left/right position of the 45° and 135° gratings varied across trials, the target stimulus in the current trial was either the same or differed by exactly 90° from the previous trial. In consequence, this study could not distinguish whether the observed bias was attractive or repulsive, which concerned both the behavioral effect and the V1 signal. Furthermore, the bias in the V1 signal was partially explained by the orientation that was presented at the same position in the previous trial, which could reflect a reactivation of the previous orientation rather than an actual altered orientation.

We have changed the Introduction accordingly.

References:

St. John-Saaltink E, Kok P, Lau HC, de Lange FP (2016) Serial Dependence in Perceptual Decisions Is Reflected in Ac6vity Pa9erns in Primary Visual Cortex. Journal of Neuroscience 36: 6186–6192.

(1b) The study motivation might also consider the findings of Ranieri et al (2022, J. Neurosci) Fornaciai, Togoli, & Bueti (2023, J. Neurosci), and Lou& Collins (2023, J. Neurosci) who all test various neural signatures of serial dependence.

Thank you. As all listed findings showed neural signatures revealing a reactivation of the previous stimulus or a response during the current trial, we have added them to the paragraph in the Introduction referring to this class of evidence for the neural basis for serial dependence.

(2) Regarding the methods and results, it would help if the initial description of the reconstruction approach, in the main text, gave more context about what data is going into reconstruction (e.g., which sensors), a more conceptual overview of what the 'reconstruction' entails, and what the fidelity metric indexes. To me, all of that is important to interpreting the figures and results. For instance, when I first read, it was unclear to me what it meant to "reconstruct the direction of S1 during the S2 epoch" (p. 10, line 199)? As in, I couldn't tell how the data/model knows which item it is reconstructing, as opposed to just reporting whatever directional information is present in the signal.(2a) Relatedly, what does "reconstruction strength" reflect in Figure 2a? Is this different than the fidelity metric? Does fidelity reflect the strength of the particular relevant direction, or does it just mean that there is a high level of any direction information in the signal? In the main text explain what reconstruction strength and what fidelity is?

Thank you for pointing this out. We applied the inverted encoding model method to MEG data from all active sensors (271) within defined time-windows of 100 ms length. MEG data was recorded in two sessions on different days. Specifically, we constructed an encoding model with 18 motion direction-selective channels. Each channel was designed to show peak sensitivity to a specific motion direction, with gradually decreasing sensitivity to less similar directions. In a training step, the encoding model was fiCed to the MEG data of one session to obtain a weight matrix that indicates how well the sensor activity can be explained by the modeled direction. In the testing step, the weight matrix was inverted and applied to the MEG data of the other session, resulting in a response profile of ‘reconstruction strengths’, i.e., how strongly each motion direction was present in a trial. When a specific motion direction was present in the MEG signal, the reconstruction strengths peaked at that specific direction and decreased with increasing direction difference. If no information was present, reconstruction strengths were comparable across all modeled directions, i.e., the response profile was flat. To integrate response profiles across trials, single trial profiles were aligned to a common center direction (i.e., 180°) and then averaged.

To quantify the accuracy of each IEM reconstruction, i.e., how well the response profile represents a specific motion direction relative to all other directions we computed the ‘reconstruction fidelity’. Fidelity was obtained by projecting the polar vector of the reconstruction at every direction angle (in steps of 1°) onto the common center (180°) and averaging across all direction angles (Rademaker et al 2019, Sprague, Ester & Serences, 2016). As such, ‘reconstruction fidelity’ is a summary metric with fidelity greater than zero indicating an accurate reconstruction.

How does the model know which direction to reconstruct? Our modelling procedure was informed about the stimulus in question during both the training and the testing step. Specifically, we informed our model during the training step about e.g., the current S2. Then, we fit the model to training data from the S2 epoch and applied it to testing data from the S2 epoch. Crucially, during the testing step the motion direction in question, i.e., current S2, becomes relevant again. For example, when S2 was 120°, the reconstructions were shifted by 60° in order to align with the common center, i.e., 180°. In addition, we also tested whether we could reconstruct the motion direction of S1 during the S2 epoch. Here, we used again the MEG data from the S2 epoch but now for S1 training. i.e., the model was informed about S1 direction. Accordingly, the recentering step during testing was done with regard to the S1 direction. Similarly, we also reconstructed the motion direction of the previous target (i.e., the previous S1 or S2), e.g., during the S2 epoch.

Together, the multi-variate pattern of MEG activity across all sensors during the S2 epoch could contain information about the currently presented direction of S2, the direction of the preceding S1 and the direction of the target stimulus from the previous trial (i.e., either previous S1 or previous S2) at the same time. An important exception from this regime was the cross-reconstruction analysis (Appendix 1—figure 2). Here we trained the encoding model on the currently relevant item (S1 during the S1 epoch, S2 during the S2 epoch and the cued item during the retro-cue epoch) of one MEG session and reconstructed the previous target on the other MEG session.

Finally, to examine shifts of the neural representation, single-trial reconstructions were assigned to two groups, those with a previous target that was oriented clockwise (CW) in relation to the currently relevant item and those with a previous target that was oriented counter-clockwise (CCW). The CCW reconstructions were flipped along the direction space, hence, a negative deviation of the maximum of the reconstruction from 180° indicated an attraction toward the previous target, whereas a positive deviation indicated a repulsion. Those reconstructions were then first averaged within each possible motion direction and then across them to account for different presentation numbers of the directions, resulting in one reconstruction per participant, epoch and time point. To examine systematic shifts, we then tested if the maximum of the reconstruction was systematically different from the common center (180°). For display purposes, we subtracted the reconstructed maximum from 180° to compute the direction shifts. A positive shift thus reflected attraction and a negative shift reflected repulsion.

We have updated the Results accordingly.

References:

Rademaker RL, Chunharas C, Serences JT (2019) Coexisting representations of sensory and mnemonic information in human visual cortex. Nature Neuroscience. 22: 1336-1344.

Sprague TC, Ester EF, Serences JT (2016) Restoring Latent Visual Working Memory Representations in Human Cortex. Neuron. 91: 694-707

(3) Then in the Methods, it would help to provide further detail still about the IEM training/testing procedure. For instance, it's not entirely clear to me whether all the analyses use the same model (i.e., all trained on stimulus encoding) or whether each epoch and timepoint is trained on the corresponding epoch and timepoint from the other session. This speaks to whether the reconstructions reflect a shared stimulus code across different conditions vs. that stimulus information about various previous and current trial items can be extracted if the model is tailored accordingly.

As reported above, our modeling procedure was informed about same stimulus during both the training and the testing step, except for the cross-reconstruction analysis.

Regarding the training and testing data, the model was always trained on data from one session and tested on data from the other session, so that each MEG session once served as the training data set and once as the test data set, hence, training and test data were independent. Importantly, training and testing was always performed in an epoch- and time point-specific way: For example, the model that was trained on the first 100-ms time bin from the S1 epoch of the first MEG session was tested on the first 100-ms time bin from the S1 epoch of the second MEG session.

Specifically, when you say "aim of the reconstruction" (p. 31, line 699), does that simply mean the reconstruction was centered in that direction (that the same data would go into reconstructing S1 or S2 in a given epoch, and what would differentiate between them is whether the reconstruction was centered to the S1 or S2 direction value)?

As reported above, during testing the reconstruction was centered at the currently relevant direction. The encoding model was trained with the direction labels of S1, S2 or the target item, corresponding to the currently relevant direction, i.e., S1 in S1 epochs, S2 in S2 epochs and target item (S1 or S2) in the retro-cue epoch. The only exception was the reconstruction of S1 during the S2 epoch. Here the encoding model was trained on the S1 direction, but with data from the S2 epoch and then applied to the S2 epoch data and recentered to the S1 direction. So here, S1 and S2 were indeed trained and tested separately for the same epoch.

(4) I think training and testing were done separately for each epoch and timepoint, but this could have important implications for interpreting the results. Namely if the models are trained and tested on different time points, and reference directions, then some will be inherently noisier than others (e.g., delay period more so than encoding), and potentially more (or differently) susceptible to bias. For instance, the S1 and S2 epochs show no attractive bias, but they may also be based on more high-fidelity training sets (i.e., encoding), and therefore less susceptible to the bias that is evident in the retrocue epoch.

Thanks for pointing this out. Training and testing were performed in an epoch- and time point-specific way. Thus, potential differences in the signal-to-noise ratio between different task phases could cause quality differences between the corresponding reconstructed MEG signals. However, we did not observe such differences. Instead, we found comparable time courses of the reconstruction fidelities and the averaged reconstruction strengths between epochs (Figure 2b and 2c, respectively). Fig. 2b, e.g., shows that reconstruction fidelity for motion direction stimuli built up slowly during the stimulus presentation, reaching its maximum only after stimulus offset. This observation may contrast to different stimulus materials with faster build-ups, like the orientation of a Gabor.

We agree with the reviewer that, regardless of the comparable but not perfectly equal reconstruction fidelities, there are good arguments to assume that the neural representation of the stimulus during its encoding is typically less noisy than during its post-encoding processing and that this difference could be one of the reasons why serial dependence emerged in our study only during the retro-cue epoch. However, the argument could also be reversed: a biased representation, which represents a small and hard-to-detect neural effect, might be easier to observe for less noisy data. So, the fact that we found a significant bias only during the potentially “noisier” retro-cue epoch makes the effect even more noteworthy.

We mentioned the limitation related to our stimulus material already at the end of the Discussion. We have now added a new paragraph to the Discussion to address the two opposing lines of reasoning.

(4) I believe the work would benefit from a further effort to reconcile these results with previous findings (i.e., those that showed repulsion, like Sheehan & Serences), potentially through additional analyses. The discussion attributes the difference in findings to the "combination of a retro-cue paradigm with the high temporal resolution of MEG," but it's unclear how that explains why various others observed repulsion (thought to happen quite early) that is not seen at any stage here. In my view, the temporal (as well as spatial) resolution of MEG could be further exploited here to better capture the early vs. late stages of processing. For instance, by separately examining earlier vs. later time points (instead of averaging across all of them), or by identifying and analyzing data in the sensors that might capture early vs. late stages of processing. Indeed, the S1 and S2 reconstructions show subtle repulsion, which might be magnified at earlier time points but then shift (toward attraction) at later time points, thereby counteracting any effect. Likewise, the S1 reconstruction becomes biased during the S2 epoch, consistent with previous observations that the SD effects grow across a WM delay. Maybe both S1 and S2 would show an attractive bias emerging during the later (delay) portion of their corresponding epoch? As is, the data nicely show that an attractive bias can be detected in the retrocue period activity, but they could still yield further specificity about when and where that bias emerges.

We are grateful for this suggestion. Before going into detail, we would like to explain our motivation for choosing the present analysis approach that included averaging time points within an epoch of interest.

Our aim was to detect a neuronal signature of serial dependence which is manifested as an attractive shift of about 3.5° degrees within the 360° direction space. To be able to detect such a small effect in the neural data and given the limited resolution of the reconstruction method and the noisy MEG signals, we needed to maximize the signal-to-noise ratio. A common method to obtain this is by averaging data points. In our study we asked subjects to perform 1022 trials, down-sampled the MEG data from the recorded sampling rate of 1200 Hz to 10 Hz (one data point per 100 ms) that we used for the estimation of reconstruction fidelity and calculated the final neural shift estimates by averaging time points that showed a robust reconstruction fidelity, thus representing interpretable data points.

Our procedure to maximize the signal-to-noise ratio was successful as we were able to reliably reconstruct the presented and remembered motion direction in all epochs (Figure 1a and 1b in the manuscript). However, the reconstruction did not work equally well for all time points within each epoch. In particular, there were time points with a non-significant reconstruction fidelity. In consequence, for the much smaller neural shift effect we did not expect to observe reliable time-resolved results, i.e., when considering each time point separately. Instead, we used the reconstruction results to define the time window in order to calculate the neural shift, i.e., we averaged across all time points with a significant reconstruction fidelity.

Author response image 1 depicts the neural shift separately for each time point during the retro-cue epoch. Importantly, the gray parts of the time courses indicate time points where the reconstruction of the presented or cued stimulus was not significant. This means that the reconstructed maxima at those time points were very variable/unreliable and therefore the neural shifts were hardly interpretable.

**Author response image 1. sa4fig1:** Time courses of the reconstruction shift reveal a tendency for an attractive bias during the retrocue phase. Time courses of the neural shift separately for each time point during the S1 (left panel), S2 (middle panel) and retro-cue epochs (right panel). Gray lines indicate time points with non-significant reconstruction fidelities and therefore very variable and non-interpretable neural reconstruction shifts. The colored parts of the lines correspond to the time periods of significant reconstruction fidelities with interpretable reconstruction shifts. Error bars indicate the middle 95% of the resampling distribution. Time points with less than 5% (equaling p < .05) of the resampling distribution below 0° are indicated by a colored circle. N = 10.

First, the time courses in the Author response image 1 show that the neural bias varied considerably between subjects, as revealed by the resampling distributions, at given time points. In this resampling procedure, we drew 10 participants in 10.000 iterations with replacement and calculated the reconstruction shift based on the mean reconstruction of the resampled participants. The observed variability stresses the necessity to average the values across all time points that showed a significant reconstruction fidelity to increase the signal-to-noise ratio.

Second, despite this high variability/low signal-to-noise ratio, Author response image 1 (right panel) shows that our choice for this procedure was sensible as it revealed a clear tendency of an attractive shift at almost all time points between 300 through 1500 ms after retro-cue onset with only a few individual time-points showing a significant effect (uncorrected for multiple comparisons). It is worth to mention that this time course did not overlap with the time course of previous target cross-reconstruction (Appendix 1—figure 2, right panel), as there was no significant target cross-reconstruction during the retro-cue epoch with an almost flat profile around zero. Also, there was no overlap with previous target decoding in the retro-cue epoch (Figure 5 in the manuscript). Here, the previous target was reactivated significantly only at early time points of 200 and 300 ms post cue onset (i.e., at time points with a non-significant reconstruction fidelity and therefore no interpretable neural shift), while the nominally highest values of the attractive neural shift were visible at later time points that also showed a significant reconstruction fidelity (Figure 2b in the manuscript).

Third, Author response image 1 (left and middle panel) shows the time courses of the neural shift during the S1 and S2 epochs. While no neural shift could be observed for S1, during the S2 epoch the time-resolved analysis indicated an initial attractive shift followed by a (nonsignificant) tendency for a repulsive shift. After averaging neural shifts across time points with a significant reconstruction fidelity, there was no significant effect with an overall tendency for repulsion, as reported in the paper. The attractive part of the neural shift during the S2 epoch was nominally strongest at very early time points (at 100-300 ms after S2 onset) and overlapped perfectly with the reactivation of the previous target as shown by the cross-reconstruction analysis (Appendix 1—figure 2, middle panel). This overlap suggests that the neural attractive shift did not reflect an actual bias of the early S2 representation, but rather a consequence of the concurrent reactivation of the previous target in the same neural code as the current representation. Finally, this neural attractive shift during S2 presentation did not correlate with the behavioral error (single trial-wise correlation: no significant time points during S2 epoch) or the behavioral bias (subject-wise correlation). In contrast, for the retro-cue epoch, we observed a significant correlation between the neural attractive shift and behavior.

Together, the time-resolved results show a clear tendency for an attractive neural bias during the retro-cue phase, thus supporting our interpretation that the attractive shift during the retro-cue phase reflects a direct neuronal signature of serial dependence. However, these additional analyses also demonstrated a large variability between participants and across time points, warranting a cautious interpretation. We conclude that our initial approach of averaging across time points was an appropriate way of reducing the high level of noise in the data and revealed the reported significant and robust attractive neural shift in the retrocue phase.

(5) A few other potentially interesting (but inessential considerations): A benchmark property of serial dependence is its feature-specificity, in that the attractive bias occurs only between current and previous stimuli that are within a certain range of similarity to each other in feature space. I would be very curious to see if the neural reconstructions manifest this principle - for instance, if one were to plot the trialwise reconstruction deviation from 0, across the full space of current-previous trial distances, as in the behavioral data. Likewise, something that is not captured by the DoG fivng approach, but which this dataset may be in a position to inform, is the commonly observed (but little understood) repulsive effect that appears when current and previous stimuli are quite distinct from each other. As in, Figure 1b shows an attractive bias for direction differences around 30 degrees, but a repulsive one for differences around 170 degrees - is there a corresponding neural signature for this component of the behavior?

We appreciate the reviewer's idea to split the data. However, given that our results strongly relied on the inclusion of all data points, i.e., including all distances in motion direction between the current S1, S2 or target and the previous target and requiring data averaging, we are concerned that our study was vastly underpowered to be able to inform whether the attractive bias occurs only within a certain range of inter-stimulus similarity. To address this important question, future studies would require neural measurements with much higher signal-to-noise-ratio than the present MEG recordings with two sessions per participant and 1022 trials in total.

**Reviewer #2 (Public Review):**
Summary:The study aims to probe the neural correlates of visual serial dependence - the phenomenon that estimates of a visual feature (here motion direction) are attracted towards the recent history of encoded and reported stimuli. The authors utilize an established retro-cue working memory task together with magnetoencephalography, which allows to probe neural representations of motion direction during encoding and retrieval (retro-cue) periods of each trial. The main finding is that neural representations of motion direction are not systematically biased during the encoding of motion stimuli, but are attracted towards the motion direction of the previous trial's target during the retrieval (retro-cue period), just prior to the behavioral response. By demonstrating a neural signature of attractive biases in working memory representations, which align with attractive behavioral biases, this study highlights the importance of post-encoding memory processes in visual serial dependence.Strengths:The main strength of the study is its elegant use of a retro-cue working memory task together with high temporal resolution MEG, enabling to probe neural representations related to stimulus encoding and working memory. The behavioral task elicits robust behavioral serial dependence and replicates previous behavioral findings by the same research group. The careful neural decoding analysis benefits from a large number of trials per participant, considering the slow-paced nature of the working memory paradigm. This is crucial in a paradigm with considerable trial-by-trial behavioral variability (serial dependence biases are typically small, relative to the overall variability in response errors). While the current study is broadly consistent with previous studies showing that attractive biases in neural responses are absent during stimulus encoding (previous studies reported repulsive biases), to my knowledge it is the first study showing attractive biases in current stimulus representations during working memory. The study also connects to previous literature showing reactivations of previous stimulus representations, although the link between reactivations and biases remains somewhat vague in the current manuscript. Together, the study reveals an interesting avenue for future studies investigating the neural basis of visual serial dependence.Weaknesses:(1) The main weakness of the current manuscript is that the authors could have done more analyses to address the concern that their neural decoding results are driven by signals related to eye movements. The authors show that participants' gaze position systematically depended on the current stimuli's motion directions, which together with previous studies on eye movement-related confounds in neural decoding justifies such a concern. The authors seek to rule out this confound by showing that the consistency of stimulus-dependent gaze position does not correlate with (a) the neural reconstruction fidelity and (b) the repulsive shift in reconstructed motion direction. However, both of these controls do not directly address the concern. If I understand correctly the metric quantifying the consistency of stimulus-dependent gaze position (Figure S3a) only considers gaze angle and not gaze amplitude. Furthermore, it does not consider gaze position as a function of continuous motion direction, but instead treats motion directions as categorical variables. Therefore, assuming an eye movement confound, it is unclear whether the gaze consistency metric should strongly correlate with neural reconstruction fidelity, or whether there are other features of eye movements (e.g., amplitude differences across participants, and tuning of gaze in the continuous space of motion directions) which would impact the relationship with neural decoding. Moreover, it is unclear whether the consistency metric, which does not consider history dependencies in eye movements, should correlate with attractive history biases in neural decoding. It would be more straightforward if the authors would attempt to (a) directly decode stimulus motion direction from x-y gaze coordinates and relate this decoding performance to neural reconstruction fidelity, and (b) investigate whether gaze coordinates themselves are history-dependent and are attracted to the average gaze position associated with the previous trials' target stimulus. If the authors could show that (b) is not the case, I would be much more convinced that their main finding is not driven by eye movement confounds.

The reviewer is correct that our eye-movement analysis approach considered gaze angle (direction) and not gaze amplitude. We considered gaze direction to be the more important feature to control for when investigating the neural basis of serial dependence that manifests, given the stimulus material used in our study, as a shift/deviation of angle/direction of a representation towards the previous target motion direction. To directly relate gaze direction and MEG data to each other we equaled the temporal resolution of the eye tracking data to match that of the MEG data. Specifically, our analysis procedure of gaze direction provided a measure indicating to which extent the variance of the gaze directions was reduced compared with random gaze direction patterns, in relation to the specific stimulus direction within each 100 ms time bin. Importantly, this procedure was able to reveal not only systematic gaze directions that were in accordance with the stimulus direction or the opposite direction, but also picked up all stimulus-related gaze directions, even if the relation differed across participants or time.

Our analysis approach was highly sensitive to detect stimulus-related gaze directions during all task phases (Appendix 1—figure 3). As expected, we found systematic gaze directions when S1 and S2 were presented on the screen, and they were reduced thereafter, indicating a clear relationship between stimulus presentation and eye movement. Systematic gaze directions were also present in the retro-cue phase where no motion direction was presented. Here they showed a clearly different temporal dynamic as compared to the S1 and S2 phases. They appeared at later time points and with a higher variability between participants, indicating that they coincided with retrieving the target motion direction from working memory.

To relate gaze directions with MEG results, we calculated Spearman rank correlations. We found that there was no systematic relationship at any time point between the stimulus related reconstruction fidelity and the amount of stimulus-related gaze direction. Even more, the correlation varied strongly from time point to time point revealing its random nature. In addition to the lack of significant correlations, we observed clearly distinct temporal profiles for gaze direction (Appendix 1—figure 3a and Appendix 1—figure 3b) and the reconstruction fidelities (Figure 2b in the manuscript, Appendix 1—figure 3c), in particular in the critical retro-cue phase.

We favored this analysis approach over one that directly decoded stimulus motion direction from x-y gaze coordinates, as we considered it hardly feasible to compute an inverted encoding model with only two eye-tracker channels as an input (in comparison to 271 MEG sensors), and to our knowledge, this has not been done before. Other decoding methods have previously been applied to x-y gaze coordinates. However, in contrast to the inverted encoding model, they did not provide a measure of the representation shift which would be crucial for our investigation of serial dependence.

We appreciate the suggestion to conduct additional analyses on eye tracking data (including different temporal and spatial resolution and different features) and their relation to MEG data. However, the first author, who ran all the analyses, has in the meantime left academia. Unfortunately, we currently do not have sufficient resources to perform additional analyses.

While the presented eye movement control analysis makes us confident that our MEG finding was not crucially driven by stimulus-related gaze directions, we agree with the reviewer that we cannot completely exclude that other eye movement-related features could have contributed to our MEG findings. However, we would like to stress that whatever that main source for the observed MEG effect was (shift of the neuronal stimulus representation, (other) features of gaze movement, or shift of the neuronal stimulus representation that leads to systematic gaze movement), our study still provided clear evidence that serial dependence emerged at a later post-encoding stage of object processing in working memory. This central finding of our study is hard to observe with behavioral measures alone and is not affected by the possible effects of eye movements.

We have slightly modified our conclusion in the Results and Appendix 1. Please see also our response to comment 1 from reviewer 3.

(2) I am not convinced by the across-participant correlation between attractive biases in neural representations and attractive behavioral biases in estimation reports. One would expect a correlation with the behavioral bias amplitude, which is not borne out. Instead, there is a correlation with behavioral bias width, but no explanation of how bias width should relate to the bias in neural representations. The authors could be more explicit in their arguments about how these metrics would be functionally related, and why there is no correlation with behavioral bias amplitude.

We are grateful for this suggestion. We correlated the individual neuronal shift with the two individual parameter fits of the behavior shift, i.e., amplitude (a) and tuning width (w). We found a significant correlation between the individual neural bias and the w parameter (r = .70, p = .0246) but not with the a parameter (r = -.35, p = .3258) during the retro-cue period (Appendix 1—figure 1). This indicates that a broader tuning width of the individual bias (as reflected by a smaller w parameter) was associated with a stronger individual neural attraction.

It is important to note that for the calculation of the neural shift, all trials entered the analysis to increase the signal-to-noise ratio, i.e., it included many trials where current and previous targets were separated by, e.g., 100° or more. These trials were unlikely to produce serial dependence. Subjects with a more broadly tuned serial dependence had more interitem differences that showed a behavioral attraction and therefore more trials affected by serial dependence that entered the calculation of the neural shift. In contrast, individual differences in the amplitude (a) parameter were most likely too small, and higher individual amplitude did not involve more trials as compared to smaller amplitude to affect the neural bias in a way to be observed in a significant correlation.

We have added this explanation to Appendix 1.

(3) The sample size (n = 10) is definitely at the lower end of sample sizes in this field. The authors collected two sessions per participant, which partly alleviates the concern. However, given that serial dependencies can be very variable across participants, I believe that future studies should aim for larger sample sizes.

We want to express our appreciation for raising this issue. We apologize that we did not explicitly explain and justifythe choice for the sample size used in our paper, in particular, as we had in fact performed a formal a-priori power analysis.

At the time of the sample size calculation, there were no comparable EEG or MEG studies to inform our power calculation. Thus, we based our calculation merely on the behavioral effect reported in the literature and, in particular, observed in a behavioral study from our lab that included four different experiments with overall more than 100 participants with 1632 trials each (see Fischer et al., 2020), in which the behavioral serial dependence effect (target vs. nontarget) was very robust. Based on the contrast between target and non-target with an effect size of 1.359 in Experiment 1, a power analysis with 80% desired power led to a small, estimated sample size of 6 subjects.

However, we expected that the detection of the neural signature of this effect would require more participants. Therefore, we based our power calculation on a much smaller behavioral effect, i.e. the modulation of serial dependence by the context-feature congruency that we observed in our previous study (Fischer et al., 2020). In particular, we focused on Experiment 1 of the previous study that used color as the feature for retro-cueing, as we planned to use exactly the same paradigm for the MEG study. In contrast to the serial dependence effect, its modulation by color resulted in a more conservative power estimate: Based on an effect size of 0.856 in that experiment, a sample size of n = 10 should yield a power of 80% with two MEG sessions per subject.

At the time when we conducted our study, two other studies were published that investigated serial dependence on the neural level. Both studies included a smaller number of data points than our study: Sheehan & Serences (2022) recorded about 840 trials in each of 6 participants, resulting in fewer data points both on the participant and on the trial level. Hajonides et al. (2023) measured 20 participants with 400 trials each, again resulting in fewer datapoints than our study (10 participants with 1022 trials each). Taken together, our a-priori sample size estimation resulted in comparable if not higher power as compared to other similar studies, making us feel confident that the estimated sample was sufficient to yield reliable results.

We have now included this description and the results of this power analysis in the Materials and Methods section.

Despite this, we fully agree with the reviewer that our study would profit from higher power. With the knowledge of the results from this study, future projects should attempt to increase substantially the signal-to-noise-ratio by increasing the number of trials in particular, in order to observe, e.g., robust time-resolved effects (see our comments to review 1).

References:

Fischer C, Czoschke S, Peters B, Rahm B, Kaiser J, Bledowski C (2020) Context information supports serial dependence of multiple visual objects across memory episodes. Nature Communication 11: 1932.

Sheehan TC, Serences JT (2022) Attractive serial dependence overcomes repulsive neuronal adaptation PLOS Biology 20: e3001711.

Hajonides JE, Van Ede F, Stokes MG, Nobre AC, Myers NE (2023) Multiple and Dissociable Effects of Sensory History on Working-Memory Performance Journal of Neuroscience 43: 2730–2740.

(4) It would have been great to see an analysis in source space. As the authors mention in their introduction, different brain areas, such as PPC, mPFC, and dlPFC have been implicated in serial biases. This begs the question of which brain areas contribute to the serial dependencies observed in the current study. For instance, it would be interesting to see whether attractive shifts in current representations and pre-stimulus reactivations of previous stimuli are evident in the same or different brain areas.

We appreciate this suggestion. As mentioned above, we currently do not have sufficient resources to perform a MEG source analysis.

**Reviewer #3 (Public Review):**
Summary:This study identifies the neural source of serial dependence in visual working memory, i.e., the phenomenon that recall from visual working memory is biased towards recently remembered but currently irrelevant stimuli. Whether this bias has a perceptual or postperceptual origin has been debated for years - the distinction is important because of its implications for the neural mechanism and ecological purpose of serial dependence. However, this is the first study to provide solid evidence based on human neuroimaging that identifies a post-perceptual memory maintenance stage as the source of the bias. The authors used multivariate pattern analysis of magnetoencephalography (MEG) data while observers remembered the direction of two moving dot stimuli. After one of the two stimuli was cued for recall, decoding of the cued motion direction re-emerged, but with a bias towards the motion direction cued on the previous trial. By contrast, decoding of the stimuli during the perceptual stage was not biased.Strengths:The strengths of the paper are its design, which uses a retrospective cue to clearly distinguish the perceptual/encoding stage from the post-perceptual/maintenance stage, and the rigour of the careful and well-powered analysis. The study benefits from high within participant power through the use of sensitive MEG recordings (compared to the more common EEG), and the decoding and neural bias analysis are done with care and sophistication, with appropriate controls to rule out confounds.Weaknesses:A minor weakness of the study is the remaining (but slight) possibility of an eye movement confound. A control analysis shows that participants make systematic eye movements that are aligned with the remembered motion direction during both the encoding and maintenance phases of the task. The authors go some way to show that this eye gaze bias seems unrelated to the decoding of MEG data, but in my opinion do not rule it out conclusively. They merely show that the strengths of the gaze bias and the strength of MEGbased decoding/neural bias are uncorrelated across the 10 participants. Therefore, this argument seems to rest on a null result from an underpowered analysis.

Our MEG as well eye-movement analysis showed that they were sensitive to pick up robustly stimulus-related effects, both for presented and remembered motion directions. When relating both signals to each other by correlating MEG reconstruction strength with gaze direction, we found a null effect, as pointed out by the reviewer. Importantly, there was also a null effect when the shift of the reconstruction (representing our main finding) was correlated with gaze direction. Furthermore, an examination of the individual time courses of gaze direction and individual MEG reconstruction strength revealed that the lack of a relationship between MEG and gaze data did not rest on a singular observation but was present across all time points. Even more, the temporal profile of the correlation varied strongly from time point to time point revealing its random nature and indicating that there was no hint of a pattern that just failed to reach significance. Taking these observations together, our MEG findings were unlikely to be explained by eye position.

Nevertheless, we agree with the reviewer that there is general problem of interpreting a null effect with a limited number of observations (and an analysis approach that focused on one out of many possible features of the gaze movement). Thus, we admit that there is a (slight) possibility that eye movements contributed to the observed MEG effects. This possibility, however, did not affect our novel finding that serial dependence occurred during the postencoding stage of object processing in working memory.

Please see also our response to point 1 from reviewer 2.

Impact:This important study contributes to the debate on serial dependence with solid evidence that biased neural representations emerge only at a relatively late post-perceptual stage, in contrast to previous behavioural studies. This finding is of broad relevance to the study of working memory, perception, and decision-making by providing key experimental evidence favouring one class of computational models of how stimulus history affects the processing of the current environment.
**Recommendations for the authors:**

**Reviewer #1 (Recommendations For The Authors):**
Minor concerns:The significance statement opens "Our perception is biased towards sensory input from the recent past." This is a semantic point, but it seems a somewhat odd statement, given there is so much debate about whether serial dependence is perceptual vs. decisional, and that the current work indeed claims that it emerges at a late, post-encoding stage.

Thank you for this point. We agree. “Visual cognition is biased towards sensory input from the recent past.” would be a more appropriate statement. According to the Journal's guidelines, however, the paragraph with the Significant Statement will be not included in the final manuscript.

It would be preferable for data and code to be available at review so that reviewers might verify some procedural points for clarity.

Code and preprocessed data used for the presented analyses are now available on OSF via http://osf.io/yjc93/. Due to storage limitations, only the preprocessed MEG data for the main IEM analyses focusing on the current direction are uploaded. For access to additional data, please contact the authors.

For instance, I could use some clarification on the trial sequence. The methods first say the direction was selected randomly, but then later say each direction occurred equally often, and there were restrictions on the relationships between current and previous trial items. So it seems it couldn't have truly been random direction selection - was the order selected randomly from a predetermined set of possibilities?

For the S1/S2 stimuli in a trial the dots moved fully coherent in a direction randomly drawn from a pool of directions between 5° and 355° spaced 10° from one another, therefore avoiding cardinal directions. Across trials, there was a predetermined set of possible differences in motion direction between the current and the previous target. This set included 18 motion direction differences, ranging from -170° to 180°, in steps of 10°. Trial sequences were balanced in a way that each of these differences occurred equally often during a MEG session.

I could also use some additional assurance the sample size (participants or data points) is sufficient for the analysis approach deployed here.

We performed a formal a-priori power analysis to justify our choice for the sample size. Please see our response to reviewer 2, point 3, where we explained the procedure of the apriori power analysis in detail. We have now included this description and the results of this power analysis in the Materials and Methods.

Did you consider a decoding approach, instead of reconstruction, to test what information predominates the signal, in an unbiased way?

Thank you for this argument. With our analysis approach based on the inverted encoding model, we believe to be unbiased, since we first reconstructed whether the MEG signal contained information about the presented and remembered motion direction. Only in the next step, we tested whether this reconstructed signal showed an offset and if so, whether this offset was biased towards or away from the previous target. A decoding approach aims to answer classification questions and is not suitable to reveal the actual shifts of the neural information. In our study, we could decode, e.g., the current direction or the previous target, but this would not answer the question of whether and at which stage of object processing the current representation was biased towards the past. Moreover, in a decoding approach to reveal which information predominates in the signal, we would have to classify different options (e.g. current information vs previous), thereby biasing the possible set of results more than in our chosen analysis.

I think the claim of a "direct" neural signature may come off as an overstatement when the spatial and temporal aspects of the attractive bias are still so coarsely specified here.

Thank you for pointing this out. We agree that the term “direct neural signature” can be seen as an overstatement when it is interpreted to indicate a narrowly defined activity of a brain region (ideally via “direct” invasive recordings) that reflects serial dependence. Our definition of the term “direct” referred to the observation of an attractive shift in a neural representation of the current target motion direction item towards the previous target. This was in contrast to previous “indirect” evidence for the neural basis of serial dependence based on either repulsive shifts of neural representations that were opposite to the attractive bias in behavior or on a reactivation of previous information in the current trial without presenting evidence for the actual neural shift. With this definition in mind, we consider the title of our study a valid description of our findings.

**Reviewer #2 (Recommendations For The Authors):**
I was wondering why the authors chose a bootstrap test for their neural bias analysis instead of a permutation test, similar to the one they used for their behavioral analysis. As far as I know, bootstrap tests do not provide guaranteed type-1 error rate control. The procedure for the permutation test would be quite straightforward here, randomly permuting the sign of each participant's neural shift and recording the group-average shift in a permutation distribution. This test seems more adequate and more consistent with the behavioral analysis.

Thank you for this comment. We adapted a resampling approach (bootstrapping) that was similar to that by Ester et al. (2020) who also investigated categorical biases and also applied a reconstruction method (Inverted Encoding Model) to assess significance of a bias of the reconstructed orientation against zero in a certain direction. The bootstrapping method relied on (a) detecting an offset against zero and (b) evaluating the robustness of the observed effect across participants. In contrast, a permutation approach, as suggested by the reviewer, assesses whether an empirical neural shift is more extreme than the permutation distribution. The permutation approach seems more suited to assess the magnitude of the shift which in our study was not a priority. Therefore, we reasoned that the bootstrapping for our inference statistics was better suited to assess the direction of the neural shift and its robustness across participants.

We have added this additional information to the Materials and Methods:

References:

Ester EF, Sprague TC, Serences JT (2020) Categorical biases in human occipitoparietal cortex. Journal of Neuroscience 40:917–931.

The manuscript could be improved by more clearly spelling how the training and testing data were labelled, particularly for the reactivation analyses. If I understood correctly, in the first reactivation analysis the authors train and test on current trial data, but label both training and testing data according to the previous trial's motion direction. In the second analysis, they label the training data according to the current motion direction, but label the testing data according to the previous motion direction. Is that correct?

Yes, this is correct. Please see also our response to reviewer 1, point 2 and 3, for a detailed description.

I was surprised to see that the shift in the reconstructed direction is about three times larger than the behavioral attraction bias. Would one not expect these to be comparable in magnitude? It would be helpful to address and discuss this in the discussion section.

Thank you for pointing this out. We agree with the reviewer that as both measures provided an identical metric (angle degree), one would expect that their magnitudes should be directly comparable. However, we speculate that these magnitudes inform only about the direction of the bias and their significant difference from zero, thus they operate on different scales and are not directly comparable. For example, Hallenbeck et al. (2022) showed that fMRI-based reconstructed orientation bias and behavioral bias correlated on both individual and group level, despite strong magnitude differences. This is in line with our observation and supports the speculation that the magnitudes of neural and behavioral biases operate on different scales and, thus, are not directly comparable.

We have updated to the Discussion accordingly.

References:

Hallenbeck GE, Sprague TC, Rahmati M, Sreenivasan KK, Curtis CE (2022) Working memory representations in visual cortex mediate distraction effects Nature Communications 12: 471.

**Reviewer #3 (Recommendations For The Authors):**
(1) It may be worth showing that the gaze bias towards the current/cued stimulus is not biased towards the previous target. One option might be to run the same analysis pipeline used for the MEG decoding but on the eye-tracking data. Another could be to remove all participants with significant gaze bias, but given the small sample size, this might not be feasible.

We appreciate this suggestion. However, as mentioned above, we currently do not have sufficient resources to conduct additional analyses on the eye tracking data.

(2) Minor typo: Figure 3c - bias should be 11.7º, not -11.7º.

Corrected. Thank you!

Note on data/code availability: The authors state that preprocessed data and analysis code will be made available on publication, but are not available yet.

Code and preprocessed data used for the present analyses are now available on OSF via http://osf.io/yjc93/. Due to storage limitations, only the preprocessed MEG data for the main IEM analyses focusing on the current direction are uploaded. For access to additional data, please contact the authors.